The EMBO Journal (2013) 32, 2751–2763
www.embojournal.org

# IFNβ-dependent increases in STAT1, STAT2, and IRF9 mediate resistance to viruses and DNA damage

**HyeonJoo Cheon[1], Elise G Holvey-Bates[1], John W Schoggins[2], Samuel Forster[3], Paul Hertzog[3], Naoko Imanaka[2], Charles M Rice[2], Mark W Jackson[4], Damian J Junk[4] and George R Stark[1,*]**

[1]Department of Molecular Genetics, Lerner Research Institute, Cleveland Clinic, Cleveland, OH, USA, [2]Laboratory of Virology and Infectious Disease, Center for the Study of Hepatitis C, Rockefeller University, New York, NY, USA, [3]Centre for Innate Immunity and Infectious Disease, Monash Institute of Medical Research, Monash University, Clayton, Victoria, Australia and [4]Department of Pathology, Case Comprehensive Cancer Center, Case Western Reserve University, Cleveland, OH, USA

A single high dose of interferon-β (IFNβ) activates powerful cellular responses, in which many anti-viral, pro-apoptotic, and anti-proliferative proteins are highly expressed. Since some of these proteins are deleterious, cells down-regulate this initial response rapidly. However, the expression of many anti-viral proteins that do no harm is sustained, prolonging a substantial part of the initial anti-viral response for days and also providing resistance to DNA damage. While the transcription factor ISGF3 (IRF9 and tyrosine-phosphorylated STATs 1 and 2) drives the first rapid response phase, the related factor un-phosphorylated ISGF3 (U-ISGF3), formed by IFNβ-induced high levels of IRF9 and STATs 1 and 2 without tyrosine phosphorylation, drives the second prolonged response. The U-ISGF3-induced anti-viral genes that show prolonged expression are driven by distinct IFN stimulated response elements (ISREs). Continuous exposure of cells to a low level of IFNβ, often seen in cancers, leads to steady-state increased expression of only the U-ISGF3-dependent proteins, with no sustained increase in other IFNβ-induced proteins, and to constitutive resistance to DNA damage.

*The EMBO Journal* (2013) **32,** 2751–2763. doi:10.1038/emboj.2013.203; Published online 24 September 2013
Subject Categories: signal transduction; microbiology & pathogens
Keywords: anti-viral genes; DNA damage resistance; interferon-β; unphosphorylated interferon-stimulated gene factor 3 (U-ISGF3)

## Introduction

Type I interferons (IFNs α and β) drive the expression of genes that encode proteins with anti-viral, anti-proliferative, pro-apoptotic, and pro-inflammatory functions. Several important

*Corresponding author. Department of Molecular Genetics, Lerner Research Institute, Cleveland Clinic, 9500 Euclid Avenue, Cleveland, OH 44195, USA. Tel.: +1 216 444 6062; Fax: +1 216 444 0512; E-mail: starkg@ccf.org

negative feedback mechanisms collaborate to terminate the expression of these genes several hours after IFN stimulation, for example, expression of the potent negative regulator SOCS1 is rapidly induced by IFN (Yoshimura *et al*, 2007). Although sustained expression of many of the initially induced proteins is deleterious to cell survival (Borden *et al*, 2007), we have discovered that the increased expression of the proteins encoded by a subset of these genes is sustained for many days (Cheon and Stark, 2009). They provide a selective advantage and cells can tolerate them.

In response to IFNβ, signal transducers and activators of transcription (STATs) are phosphorylated on C-terminal tyrosine residues (Y701 for STAT1 and Y690 for STAT2), followed by their combination with IFN response factor 9 (IRF9) to form the tripartite transcription factor IFN-stimulated gene factor 3 (ISGF3), which drives the expression of >100 IFNβ-stimulated genes (ISGs, Borden *et al*, 2007). Previously, we showed that STAT1 lacking phosphorylation of Y701 (U-STAT1) sustains the response to IFNβ for several days (Cheon and Stark, 2009) and, even in the absence of IFNβ, high levels of U-STAT1 induced the expression of a subset of ISGs, including *IFI27, BST2, OAS1, OAS2, OAS3,* and *STAT1* itself. Many of the encoded proteins have anti-viral activities (Dong *et al*, 2004; Borden *et al*, 2007; Neil *et al*, 2008; Randall and Goodbourn, 2008; Sadler and Williams, 2008; Tan *et al*, 2008; Brass *et al*, 2009; Itsui *et al*, 2009; Schmeisser *et al*, 2010; Tang *et al*, 2010; Miyashita *et al*, 2011; Schoggins *et al*, 2011; Oudshoorn *et al*, 2012). We have now elucidated the mechanism and the additional biological consequences of U-STAT1-induced gene expression, finding that IFNβ also induces the expression of un-phosphorylated STAT2 (U-STAT2) and IRF9, which combine with U-STAT1 to form un-phosphorylated ISGF3 (U-ISGF3), a novel transcription factor in which these proteins form a ternary complex without tyrosine phosphorylation. U-ISGF3 in turn maintains the expression of a subset of the initially induced ISGs whose protein products lead to extended resistance to virus infection and DNA damage. Interestingly, expression of the same subset of ISGs is uniquely increased in radiation-resistant cancer cells (Khodarev *et al*, 2004; Cheon *et al*, 2011), in cancer cells resistant to a variety of DNA damaging treatments (Gongora *et al*, 2008; Luszczeck *et al*, 2010), and in cancer cells from glioblastoma and breast cancer patients who responded poorly to chemo- or radiation therapy (Weichselbaum *et al*, 2008; Duarte *et al*, 2012). We show that prolonged exposure of cells to a low level of IFNβ induces a steady state in which only the U-ISGF3-dependent genes are expressed, suggesting that secretion of IFNβ by cancer cells may account for their similar phenotype.

## Results

### The expression of anti-viral genes is sustained for several days after IFNβ treatment, along with increased levels of STAT1, STAT2, and IRF9 proteins

Over a hundred genes are induced by IFNβ quickly, in response to the tyrosine phosphorylation of STATs 1 and 2

and subsequent formation of ISGF3, but the expression of many genes is downregulated as the level of ISGF3 decreases. However, the expression of many anti-viral genes that are induced initially by IFNβ is sustained and even increased by increased expression of U-STAT1, the levels of which remain high for many days (Cheon and Stark, 2009). As shown in Figure 1A, the expression of four representative anti-viral genes (*IFI27*, *BST2*, *OAS2*, and *MX1*) is induced greatly after 24 h and sustained at high levels for at least 72 h after a single treatment with IFNβ (3 IU/ml) of two different human non-cancer cell lines, hTERT-HME1 mammary epithelial cells and BJ fibroblasts. To test whether the continued expression of these genes might be due to the presence of a low residual level of phosphorylated STATs 1 and 2, we examined STAT expression and phosphorylation in response to a much higher concentration of IFNβ (50 IU/ml). Even at this high concentration, phosphorylated STATs 1 and 2 were seen only transiently, and we detected little phosphorylated STAT1 or phosphorylated STAT2 after 48 h (Figure 1B). However, the expression of STAT1, STAT2, and IRF9 was increased greatly after 24 h and was sustained for at least 72 h, with kinetics similar to the kinetics of anti-viral gene expression shown in Figure 1A. In contrast to the sustained expression of the four anti-viral genes noted above, the expression of IRF1, an ISG whose expression is driven by ISGF3 and not by U-STAT1, increased transiently and decreased in parallel with the levels of phosphorylated STATs 1 and 2 (Figure 1B), showing that, even if phosphorylated ISGF3 was still present at levels below our ability to detect it, there was not enough to drive the expression of this target gene. The increased levels of STAT1, STAT2, IRF9, and several U-STAT1-induced genes (*IFI27*, *OAS2*, and *MX1*) lasted for at least 12 days after a single treatment with IFNβ (50 IU/ml), while the expression of ISGs (*MYD88*, *IRF1*, and *IFI16*) that are not induced by U-STAT1 (Cheon and Stark, 2009) returned to basal levels after 3 days or sooner (Supplementary Figure S1). In contrast to their prolonged expressions in BJ and hTERT-HME1 cells, the expression of *IFI27*, *OAS2*, and *MX1* was transient in AG14412 umbilical cord fibroblasts, where IFNβ induced the tyrosine phosphorylation of STATs 1 and 2, did not increase the expression of STAT1 and IRF9 proteins, and increased STAT2 protein expression minimally (Figure 1C). We observed a similarly transient expression of *IFI27*, *OAS2*, and *MX1* in STAT1-null fibroblasts reconstituted with wild-type STAT1 (Figure 1D). Since STAT1 gene expression in these cells is regulated by the CMV promoter in the vector and not by the endogeneous STAT1 promoter, STAT1 protein expression is not increased in response to IFNβ. These results show that increased levels of STAT1, STAT2, and IRF9 are likely to be important for the prolonged expression of anti-viral genes, while tyrosine phosphorylation of STATs 1 and 2 is important for initial gene expression.

### High levels of U-STAT1, U-STAT2, and IRF9 are necessary and sufficient for the induction of some anti-viral genes

Our previous microarray analysis showed that increased expression of either wild-type or Y701F mutant STAT1 led to increased expression of 30 genes without IFN stimulation in BJ cells, which already express substantial amounts of STAT2 and IRF9, but not in hTERT-HME1 cells, which express little STAT2 and IRF9 (Cheon and Stark, 2009), indicating that

U-STAT1 may not induce gene expression without a sufficient amount of STAT2 and IRF9. To investigate the role of STAT2 and IRF9, we stably increased the expression of STAT2 and IRF9, together with STAT1, in hTERT-HME1 cells. Western analyses confirmed that, in the absence of treatment with IFNβ, the highly expressed STATs were not phosphorylated on their tyrosine residues (Figure 2A). The expression of *IFI27*, *OAS2*, and *MX1* was very low in control hTERT-HME1 cells (Figure 2B, column 1) and their expression was not increased when the levels of U-STAT1, U-STAT2, or IRF9 were increased one at a time without treatment with IFNβ (Figure 2B, columns 2–4). Furthermore, the combined high expression of U-STAT1 plus U-STAT2 or U-STAT1 plus IRF9 still did not increase the expression of these genes in hTERT-HME1 cells (Figure 2B, columns 5 and 6). However, the expression of the target genes increased strongly when the levels of U-STAT2 and IRF9 were increased together (column 7) and increased even more when U-STAT1 expression, already significant, was increased further (column 8). Y701F-STAT1 is a dominant negative protein because it binds to IFN receptor SH2 domains but cannot be phosphorylated on Y701, thus blocking access of the wild-type protein to the receptor. However, high expression of Y701F-STAT1, U-STAT2, and IRF9 together in hTERT-HME1 cells significantly increased the expression of the target genes, *IFI27*, *OAS1*, *OAS2*, *MX1*, *IFIT1*, and *IFIT3* (Figure 2C), indicating that STAT1 tyrosine phosphorylation was not involved. In contrast, the expression of additional ISGs (*MYD88*, *ADAR*, *IFI16*, and *IRF1*), which are induced transiently by ISGF3 but not sustained at late times after IFN treatment, was not increased by higher levels of Y701F-STAT1, U-STAT2, and IRF9 (Figure 2D). To exclude the possibility that phosphorylation of endogeneous wild-type STAT1 is involved in the expression of the genes, we also used STAT1-null fibroblasts reconstituted with Y701F-STAT1, where STAT1 cannot be phosphorylated on residue Y701. While IFNβ does not increase the expression of *IFI27*, *OAS2*, and *MX1* in these cells (Supplementary Figure S2), high expression of STAT2 and IRF9 together with Y701F-STAT1 readily increased those three ISGs (Figure 2E), but not the transiently induced ISGs *MYD88*, *IFI16*, and *IRF1* (Figure 2F).

### High levels of U-STAT1, U-STAT2, and IRF9 protect cells from virus infection

Vesicular stomatitis virus (VSV, a negative ssRNA virus) was less infectious in hTERT-HME1 cells expressing high levels of wild-type STAT1 (WT) or Y701F-STAT1 (YF), together with U-STAT2 and IRF9, than in control cells (Vec, Figure 3A, left panel; Supplementary Figure S3A). The titres of infectious VSV were reduced by high levels of wild-type STAT1/STAT2/IRF9 (WT) or Y701F-STAT1/STAT2/IRF9 (YF) by 51-fold or 33-fold, respectively. In cells overexpressing wild-type STAT1/STAT2/IRF9 (WT), virus replication was inhibited more efficiently in the presence of IFNβ (Supplementary Figure S3B), because increased levels of ISGF3 formed by wild-type STAT1/STAT2/IRF9 sensitize cells to IFNs. However, anti-viral effects in cells overexpressing Y701F-STAT1/STAT2/IRF9 (YF) were not influenced by IFNβ in the media (Supplementary Figure S3B), showing that the Y701F-STAT1/STAT2/IRF9-induced anti-viral effects resulted solely from the high levels of U-STAT1, U-STAT2, and IRF9 proteins rather than the IFN-induced phosphorylation of STATs 1 and 2.

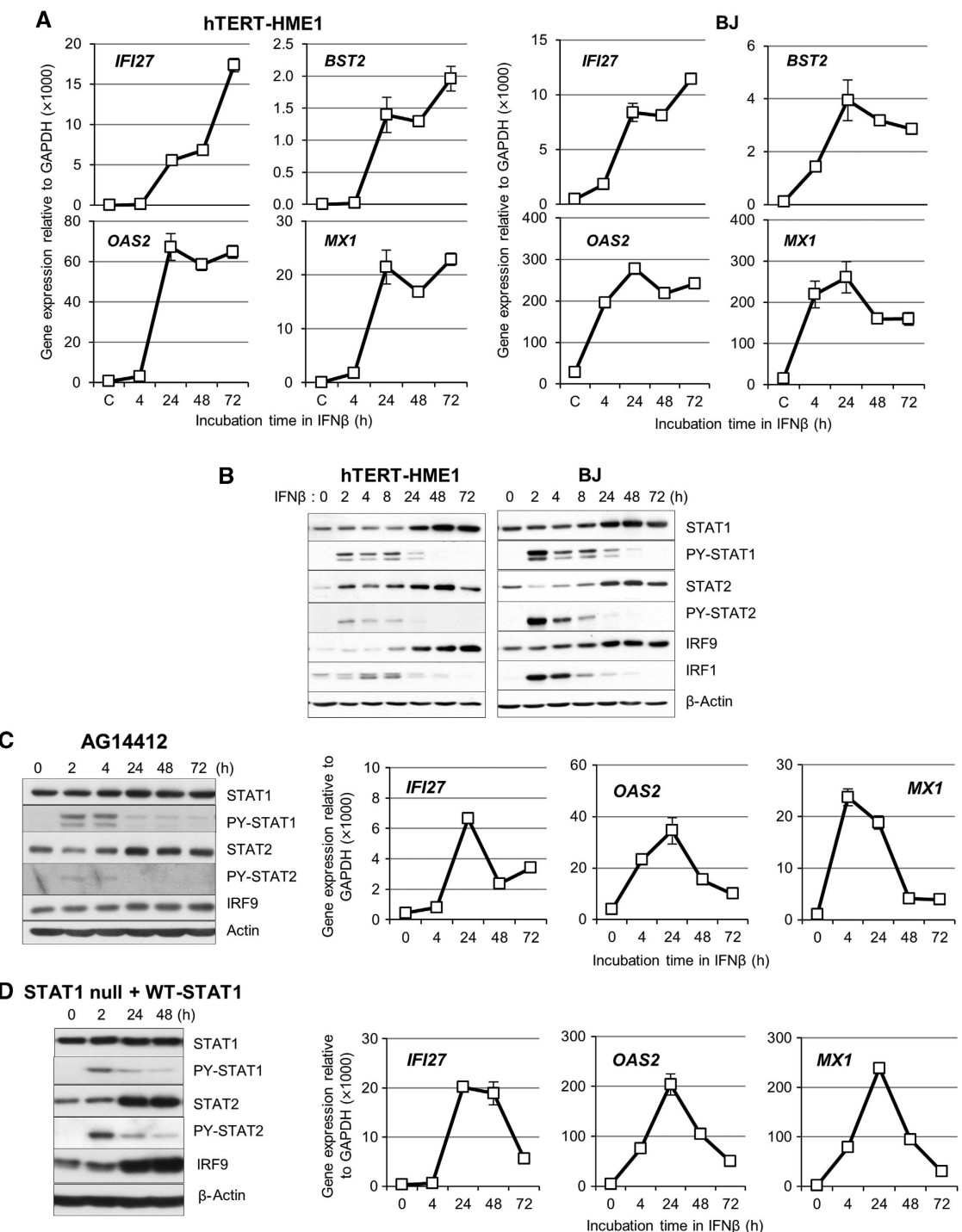

**Figure 1** The expression of anti-viral genes is sustained for several days after IFN stimulation, along with increased levels of the STAT1, STAT2, and IRF9 proteins. (**A**) The expression of the anti-viral genes *IFI27*, *BST2*, *OAS2*, and *MX1* was analysed by real-time PCR after stimulation of hTERT-HME1 or BJ cells with IFNβ (3 IU/ml). The levels of gene expression were calculated semi-quantitatively by using the ΔΔCt method (see Materials and methods). The data are represented as means of triplicate PCR analyses ± standard deviations (s.d.). (**B**) hTERT-HME1 or BJ cells were treated with IFNβ (50 IU/ml) and the levels of total IRF9 and STATs 1 and 2, or tyrosine-phosphorylated STATs 1 and 2 (PY-701-STAT1 or PY-690-STAT2) were analysed by the western method. (**C, D**) AG14412 umbilical fibroblasts (**C**) and STAT1-null fibroblasts transfected with the lentiviral vector containing wild-type STAT1 (**D**) were used. Left, Cells were treated with IFNβ (50 IU/ml) and the levels of total IRF9, STAT1, and STAT2, or tyrosine-phosphorylated STATs (PY-701-STAT1 or PY-690-STAT2) were analysed by the western method. Right, the expression of the *IFI27*, *OAS2*, and *MX1* genes was analysed by real-time PCR after stimulation with IFNβ (3 IU/ml). The levels of gene expression were calculated semi-quantitatively by using the ΔΔCt method. The data are represented as means of triplicate PCR analyses ± s.d. Source data for this figure is available on the online supplementary information page.

The replication of encephalomyocarditis virus (EMCV, a positive ssRNA virus) was also inhibited, five-fold by high levels of wild-type STAT1/STAT2/IRF9 or four-fold by

Y701F-STAT1/STAT2/IRF9, when assayed 6 h after infection (Figure 3A, right panel). We also examined the effect of high levels of wild-type- or Y701F-STAT1/STAT2/IRF9 after several

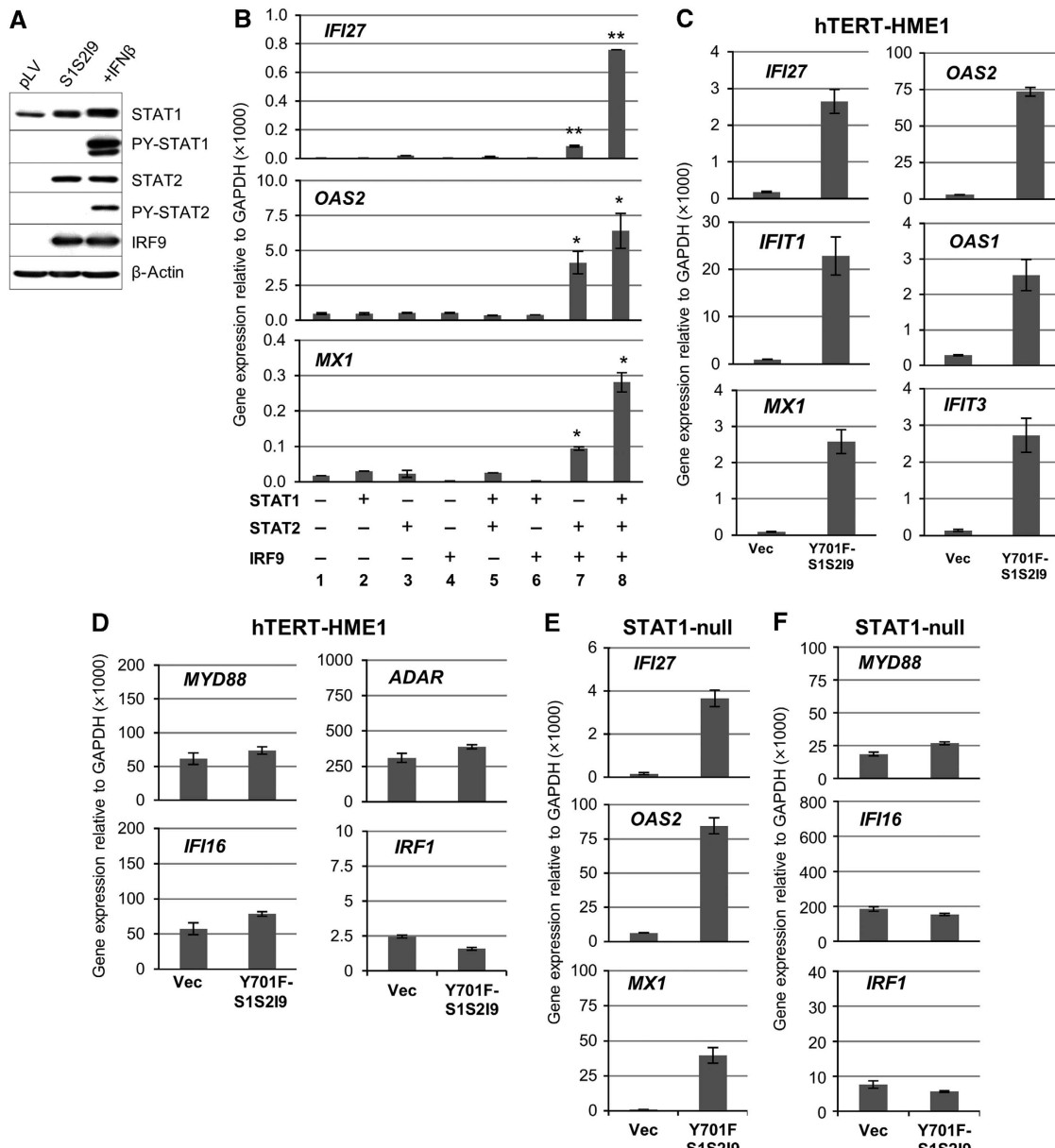

**Figure 2** High levels of STAT1, STAT2, and IRF9 proteins are necessary and sufficient for the induction of some anti-viral genes without IFN-induced phosphorylation. (**A**) The phosphorylation status of STAT1 (Y701) and STAT2 (Y690) was examined in hTERT-HME1 cells transfected with empty pLV vector or pLV-STAT1, -STAT2, and -IRF9 together (S1S2I9). As a positive control, S1S2I9 cells were treated with 500 IU/ml of IFNβ for 1 h. (**B**) The expression of the indicated genes was measured by real-time PCR in hTERT-HME1 cells stably transfected with various combinations of STAT1, STAT2, and IRF9 in the lentiviral vector pLV. The levels of gene expression were calculated semi-quantitatively by using the ΔΔCt method. The data are represented as means of triplicate PCR analyses ± s.d. ** represents $P<0.01$ and * represents $P<0.05$ by two-tailed *t*-test, compared to vector-transfected controls (column 1). (**C**, **D**) The expression of the indicated genes was measured by real-time PCR in hTERT-HME1 cells stably transfected with empty pLV vector (Vec), or pLV-Y701F-STAT1, pLV-STAT2, and pLV-IRF9 together (YF-S1S2I9). The levels of gene expression were calculated semi-quantitatively by using the ΔΔCt method. The data are represented as means of triplicate PCR analyses ± s.d. (**E**, **F**) STAT1-null fibroblasts were stably transfected with empty pLV vector (Vec), or pLV-Y701F-STAT1, pLV-STAT2, and pLV-IRF9 together (YF-S1S2I9). The expression of the indicated genes was measured by real-time PCR. The levels of gene expression were calculated semi-quantitatively by using the ΔΔCt method. The data are represented as means of triplicate PCR analyses ± s.d. Source data for this figure is available on the online supplementary information page.

cycles of virus replication. Infected cells are eventually lysed by VSV and EMCV, and we measured the surviving cells after 48 h. hTERT-HME1 cells were infected with $10–10^{-5}$ multiplicity of infection (MOI) of VSV or EMCV (Figure 3B). Control cells (Vec) were completely killed at $10^{-4}$ MOI of VSV or $10^{-2}$ MOI of EMCV, but wild-type STAT1/STAT2/IRF9-transfected cells were 100 times more resistant to VSV and >1000 times more resistant to EMCV in this assay.

Y701F-STAT1/STAT2/IRF9-transfected cells showed similar levels of resistance, confirming that the anti-viral effects were induced by high levels of U-STAT1, U-STAT2, and IRF9 proteins independently of virally induced IFN stimulation. We tested additional RNA viruses by infecting the above cells with GFP-tagged VSV, parainfluenza virus type 3 (PIV3), or yellow fever virus (YFV). Consistently, we confirmed using an alternative method, FACS analysis, that high levels of

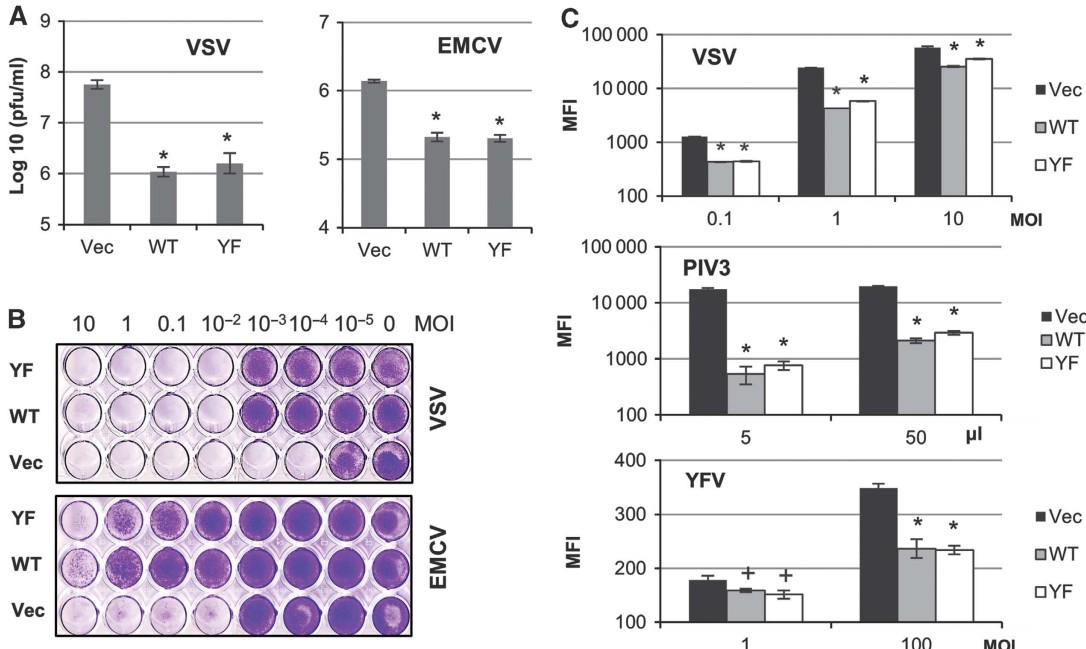

**Figure 3** High levels of STAT1, STAT2, and IRF9 proteins protect cells from various RNA viruses in an IFN-independent manner. hTERT-HME1 cells transfected with empty vector (Vec), wild-type STAT1/STAT2/IRF9 (WT) or Y701F-STAT1/STAT2/IRF9 (YF) were used. (**A**) Cells were infected with 0.1 MOI of VSV or 1 MOI of EMCV, and the infected cells and cell-culture media were collected after 10 h (VSV) or 6 h (EMCV). The infectious viral titres in the collected samples were analysed by plaque assays on Vero cells. The data are represented as means of triplicate infections ± s.d. An asterisk (*) represents $P<0.01$, by two-tailed *t*-test, compared to cells transfected with empty vector (Vec). (**B**) Cells were infected with VSV or EMCV ($10$–$10^{-5}$ MOI). After 48 h, the surviving cells were fixed with methanol (VSV) or 4% paraformaldehyde (EMCV) and stained with crystal violet. (**C**) Cells were infected with recombinant viral constructs (VSV, PIV3, or YFV) expressing GFP. After 8 h (VSV) or 48 h (PIV3 or YFV), GFP fluorescence was monitored by FACS analyses. The data are represented as mean GFP intensities (MFI) ± s.d. of triplicate infections. An asterisk (*) represents $P<0.01$ and a cross (+) represents $P<0.05$, by two-tailed *t*-test, compared to Vec cells.

wild-type- or Y701F-STAT1/STAT2/IRF9 reduced VSV replication after 8 h (Figure 3C, $P<0.01$). The replication of PIV3 (a negative ssRNA virus) was also inhibited significantly by high levels of U-STAT1/U-STAT2/IRF9, by >10-fold 48 h after infection ($P<0.01$). High levels of U-STAT1/U-STAT2/IRF9 also inhibited significantly the replication of YFV (a positive ssRNA virus), by 30% 48 h after infection with 100 MOI of virus ($P<0.01$). In summary, our results show that increased levels of U-STAT1, U-STAT2, and IRF9 are able to inhibit infection by several different RNA viruses without IFN treatment.

### U-STAT1, U-STAT2, and IRF9 form U-ISGF3, which binds to IFN stimulated response elements in target gene promoters

We examined whether U-STAT1, U-STAT2, and IRF9 could form a complex without phosphorylation, using co-immuno-precipitation (Co-IP) from hTERT-HME1 cells expressing high levels of these proteins (Figure 4A). Since the interaction between STAT1 and IRF9 in classical ISGF3 was reported to be unstable (Martinez-Moczygemba *et al*, 1997), we used the cleavable cross-linking reagent dimethyl-3,3′-dithiobis-propinimidate (DTBP). DTBP did stabilize the interaction between U-STAT1 and IRF9 (Figure 4A, lane 2), but we were still able to observe this interaction without cross-linking in the nuclear fractions of hTERT-HME1 cells expressing high levels of U-STAT1, U-STAT2, and IRF9 (lanes 3 and 7). The interactions between STAT1 and STAT2 (lanes 2, 3, and 6) and between STAT2 and IRF9 (lanes 6 and 7) were clearly observed in the nuclear fractions. We performed chromatin-

immunoprecipitation (ChIP) assays using hTERT-HME1 cells expressing high levels of U-STAT1, U-STAT2, and IRF9 in the absence of IFN treatment. Sheared chromatin (<1 Kb) was precipitated with antibodies against STAT1, STAT2, or IRF9, and the DNAs were amplified by real-time PCR, using primers spanning the most highly conserved IFN stimulated response elements (ISREs) (striped triangles in Supplementary Figure S4A) in each promoter, identified by using the transcription factor search program TFSEARCH (http://www.cbrc.jp/research/db/TFSEARCH.html). The IRF9 antibody enriched ISRE-containing promoter regions of the *IFI27*, *OAS2*, and *MX1* genes, by about 3.5-fold, compared to non-immune IgG (Figure 4B, upper panel, $P<0.05$). Analysis with an STAT1 antibody also showed enhanced binding to the ISREs of the *IFI27*, *OAS2*, and *MX1* genes, by about three-fold (Figure 4B, upper panel, $P<0.05$). STAT2 also bound to the same ISREs, with an enrichment of four- to five-fold (Figure 4B, lower panel, $P<0.05$). Promoter occupancy by U-ISGF3 was not observed in control cells transfected with empty vector (Supplementary Figure S4B). We conclude that the amount of the ternary U-ISGF3 complex is increased in response to high levels of U-STAT1, U-STAT2, and IRF9 without IFN-induced phosphorylation and is present on ISREs in the promoters of U-ISGF3 target genes. However, U-ISGF3 did not bind to ISREs in the promoters of ISGs (*MYD88*, *IRF1*, and *ADAR*) that are not induced by U-ISGF3 (Figure 4C).

### U-ISGF3-induced genes have distinct ISREs

We classified genes into U-ISGF3-induced genes and classical ISGF3-induced genes using microarray data. We identified

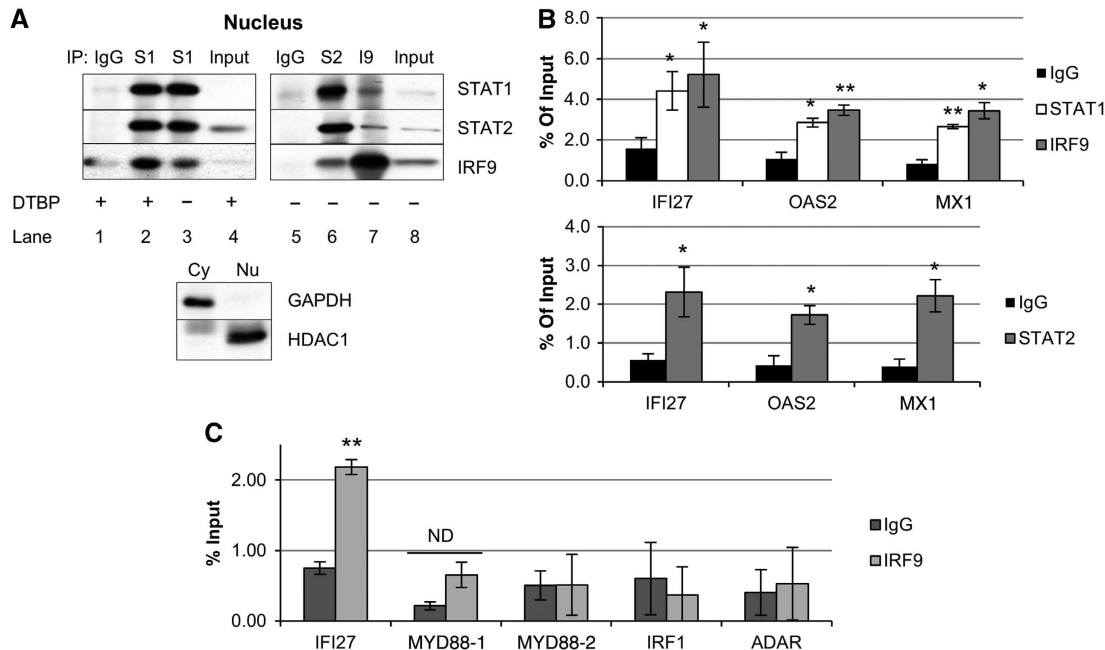

**Figure 4** U-STAT1, U-STAT2, and IRF9 form U-ISGF3, which binds to ISREs on the target gene promoters. hTERT-HME1 cells expressing high levels of U-STAT1, U-STAT2, and IRF9 without IFN stimulation were analysed by co-immunoprecipitation (Co-IP) and chromatin-immunoprecipitation (ChIP) assays. (**A**) Nuclear proteins were used for Co-IP with normal rabbit IgG or rabbit polyclonal antibodies against STAT1, STAT2, or IRF9. To stabilize protein–protein interactions, the nuclear fraction was treated with the cleavable cross-linker, dimethyl-3,3′-dithiobis-propinimidate (DTBP, lanes 1, 2, and 4). Mouse monoclonal antibodies against STAT1, STAT2, or IRF9 were used for the western method. The purity of the fractions was assessed by determining the levels of GAPDH (a cytoplasmic protein) and HDAC1 (a nuclear protein) in the input lysates by the western method. (**B**, **C**) Total protein lysates were cross-linked with 1% formaldehyde and the cell lysates were cross-linked with DTBP. Chromatin was sheared into <1 kb lengths by sonication. Rabbit polyclonal antibodies against STAT1, STAT2, or IRF9, or comparable amounts of normal rabbit IgG, were used for immunoprecipitations. Real-time PCR was performed to amplify the precipitated DNAs with primer pairs spanning ISREs in the promoters of *IFI27, OAS2, MX1, MYD88, IRF1,* and *ADAR* genes. *MYD88-1* and *-2* mean 2 different ISREs in the promoter of *MYD88* gene. The amount of amplified DNA was calculated by using the standard curve method. The values (% input) are the percentages of DNA amount in immunoprecipitated samples compared to 2% input DNA. The data are represented as means of triplicate PCR analyses ± s.d. ** represents $P < 0.01$ and * represents $P < 0.05$, by two-tailed *t*-test, compared to the IgG control. ND, not different statistically ($P > 0.05$, by two-tailed *t*-test). Source data for this figure is available on the online supplementary information page.

150 genes that are upregulated by IFNβ after 6 h, indicating that these genes are likely to have ISREs in their promoters. Indeed, analysis of the structures of transcription factor binding sites revealed a significant enrichment of canonical ISREs within the 150 IFNβ-induced genes compared with putative sites identified in all genes ($P < 0.001$). Among these IFNβ-induced genes, only 29 (20%) were induced by the upregulation of Y701F-STAT1, and we assumed that these are induced by U-ISGF3. Morrow *et al* (2011) reported that IFNγ induces the expression of anti-viral genes through another form of ISGF3, consisting of PY-STAT1, U-STAT2, and IRF9. Among the remaining 121 IFNβ-induced genes (150 IFNβ-induced genes minus 29 U-ISGF3-induced genes), 73 are induced by IFNγ (6 h). We excluded those genes, assuming that they have ISREs different from those of the genes induced only by IFNβ (48 genes, listed in Supplementary Table S1). To identify differences in ISREs, a guided analysis was performed on the genes induced only by classical ISGF3 and the genes induced by U-ISGF3 (Figure 5A). Genes induced by ISGF3 but not by U-ISGF3 contain ISREs similar to the canonical Transfac-annotated ISRE site. The U-ISGF3-induced genes also have canonical ISRE sites, which have additional conserved sequences in the 5′ and 3′ flanking regions (Figure 5B). The conserved ISRE sequences of the two groups are statistically different, applying symmetrized, position-averaged Kullback-Leibler

distance ($P < 0.05$). These results suggest that all IFNβ-induced genes are transcribed when classical ISGF3 binds to canonical ISREs at early times, but that only a subset of these genes, which contain ISREs with variant flanking sequences, can be further induced by U-ISGF3 at late times.

### U-ISGF3 induces resistance to DNA damage

Khodarev *et al* (2004) found that some of the IFN-induced genes were upregulated in radiation-resistant cancer cells compared to sensitive cells. Intriguingly, the IFN-induced genes related to DNA damage resistance are exclusively U-ISGF3-induced genes (Cheon *et al*, 2011). No other ISGs described here as ISGF3-induced genes, such as *ADAR*, *IRF1*, and *IFI16*, were elevated in DNA damage-resistant cells. The IFN-related DNA damage-resistant genes are marked with crosses (+) in Figure 5A.

The expression of the STAT1, STAT2, and IRF9 proteins varies widely in different cell types, as shown here by comparing normal human mammary epithelial cells to normal human fibroblasts. Their expression levels are also different in small cell lung carcinoma (SCLC) cell lines established from different individuals (Figure 6A). When these SCLC cell lines were treated with doxorubicin, etoposide, or ionizing radiation, there was a correlation between the levels of U-STAT1, U-STAT2, or IRF9 and cell survival (Figure 6B). SCLC lines expressing high levels of STAT1,

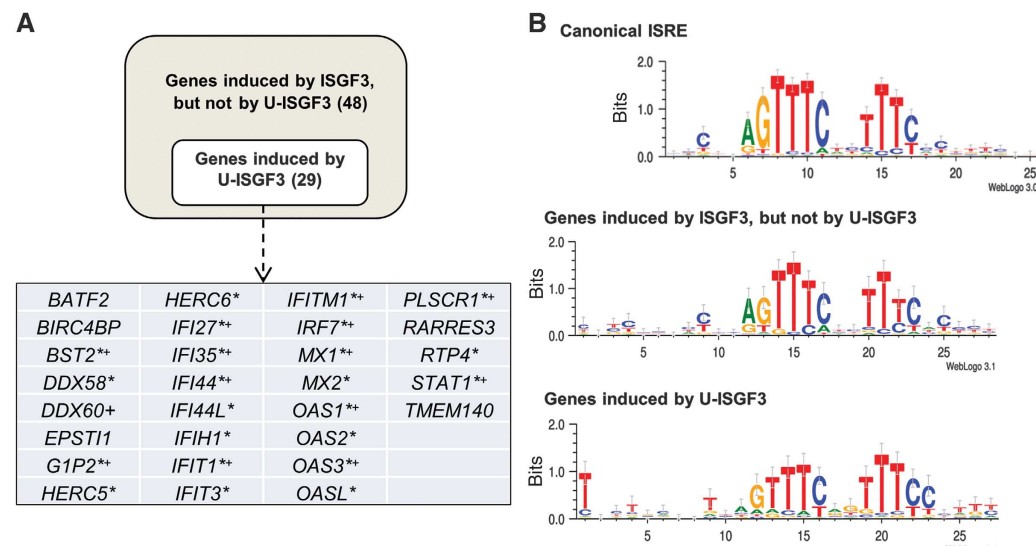

**Figure 5** U-ISGF3-induced genes have distinct ISREs. (**A**) BJ cells were treated with 3 IU/ml of IFNβ for 6 h, followed by microarray analyses for the expression of IFNβ-induced genes. To identify the U-ISGF3-induced genes, the cells were stably transfected with a lentiviral vector encoding Y701F-STAT1, without IFN treatment. The analysis criteria are described in 'Materials and methods'. The genes marked with asterisks (*) are those with known anti-viral functions, and those marked with crosses (+) are known to be upregulated in DNA damage-resistant cancer cells. *DDX58* is also known as *RIG-I*, and *IFIH1* is known as *MDA5*. *DDX60* and *TMEM140* are newer names for *FLJ20035* and *FLJ11000*, respectively. (**B**) The ISRE sites in promoter regions (2500 base pairs upstream and 500 base pair downstream from the annotated transcription start sites) of 48 genes induced by ISGF3 and not by U-ISGF3 (listed in Supplementary Table S1), and those of 29 genes induced by U-ISGF3 were analysed and the results were visualized using the WebLogo software. The canonical ISRE shown in the top panel is annotated in Transfac as a standard sequence. The conserved ISRE sequences of the U-ISGF3-induced genes (bottom panel) are statistically different from the sequences of the genes induced by ISGF3 but not by U-ISGF3 (middle panel), applying symmetrized, position-averaged Kullback-Leibler distance ($P < 0.05$).

STAT2, and IRF9 (H196 and H2195) are much more resistant to DNA damage. Since these cells have a low constitutive level of PY-STAT1, the high expression of STAT1, STAT2, and IRF9 might be due to the constitutive production of low levels of IFN. We also observed that the loss of p53 increased the expression of STAT1 (Supplementary Figure S5), which can be explained by the recent finding of Leonova *et al* (2013) that p53 helps to regulate the expression of dsRNA in cells, resulting in increased secretion of type I IFNs when p53 is not active. We postulated that chronic exposure to a low concentration of IFN might lead to a steady state in which the levels of IRF9, U-STAT1, and U-STAT2 were increased and in which the tyrosine phosphorylation of STATs 1 and 2 had been downregulated by negative regulators, leading to sustained U-ISGF3-induced gene expression. To test this idea, we treated BJ cells with 0.5 IU/ml of IFNβ every other day for 16 days. As expected, brief exposure to IFNβ induced the phosphorylation of STAT1 after 2 h (Figure 6C, right panel). However, repeated exposure to a low concentration of IFNβ increased the levels of STAT1, STAT2, and IRF9 without prolonged tyrosine phosphorylation of either STAT1 or STAT2 (Figure 6C, left panel). There was also a marked increase in the expression of the U-ISGF3-induced genes *IFI27*, *BST2*, *OAS2*, *MX1*, *IFIT1*, and *IFIT3* (Figure 6D), but not *MYD88*, *IFI16*, *ADAR* and *IRF1*, ISGs that are induced by ISGF3 but not by U-ISGF3 (Figure 6E). We conclude that continuous exposure of cells to low levels of IFNβ leads to persistent steady-state expression of only the U-ISGF3-dependent subset of ISGs, along with increased levels of STAT1, STAT2, and IRF9, independently of tyrosine-phosphorylated STATs 1 and 2. When the level of U-ISGF3 was decreased by knocking U-STAT1 down in BJ cells, the reduced STAT1 expression led to increased sensitivity to doxorubicin

(Figure 6F). shRNAs against STAT1 and IRF9 also increased the sensitivity to doxorubicin in the H196 SCLC cell line (Supplementary Figure S6). These results show that high levels of U-ISGF3 increase resistance to DNA damage as well as resistance to virus infections.

## Discussion

Figure 7A describes our working model of how anti-viral effects are prolonged after a single exposure to high levels of IFNβ. For a quick initial response, classical ISGF3, a complex of IRF9 and tyrosine phosphorylated STATs 1 and 2, mediates the induction of many ISGs, including *STAT1*, *STAT2*, and *IRF9*. As the levels of phosphorylated STATs are decreased during the course of a few hours, the expression of the ISGF3 target genes that are induced initially (e.g., *IRF1*, *ADAR*, and *MYD88*) decreases in parallel. At late times after IFN stimulation, the high levels of IRF9 and tyrosine un-phosphorylated STATs 1 and 2 proteins greatly increase the amount of U-ISGF3 and its target genes, a subset of ISGs (*IFI27*, *OAS2*, *MX1*, *BST2*, *IFIT1*, and *IFIT3*), which were previously found to be induced by U-STAT1 (Cheon and Stark, 2009). The ISRE of IFI27 is occupied by IRF9 but not by phosphorylated STAT1 72 h after IFNβ stimulation (Supplementary Figure S4C), indicating that U-ISGF3, not classical ISGF3, binds to ISREs of target ISGs at late stages. The system driven by U-ISGF3 sustains, for at least 12 days, the expression of those ISGs whose protein products are not harmful to cells but still provide a significant level of anti-viral protection, helping to clear viruses more completely and over a long period of time. We have described variations in the ISREs of U-ISGF3-induced genes compared to those of genes induced only by ISGF3 (Figure 5). However, it is possible that other factors in

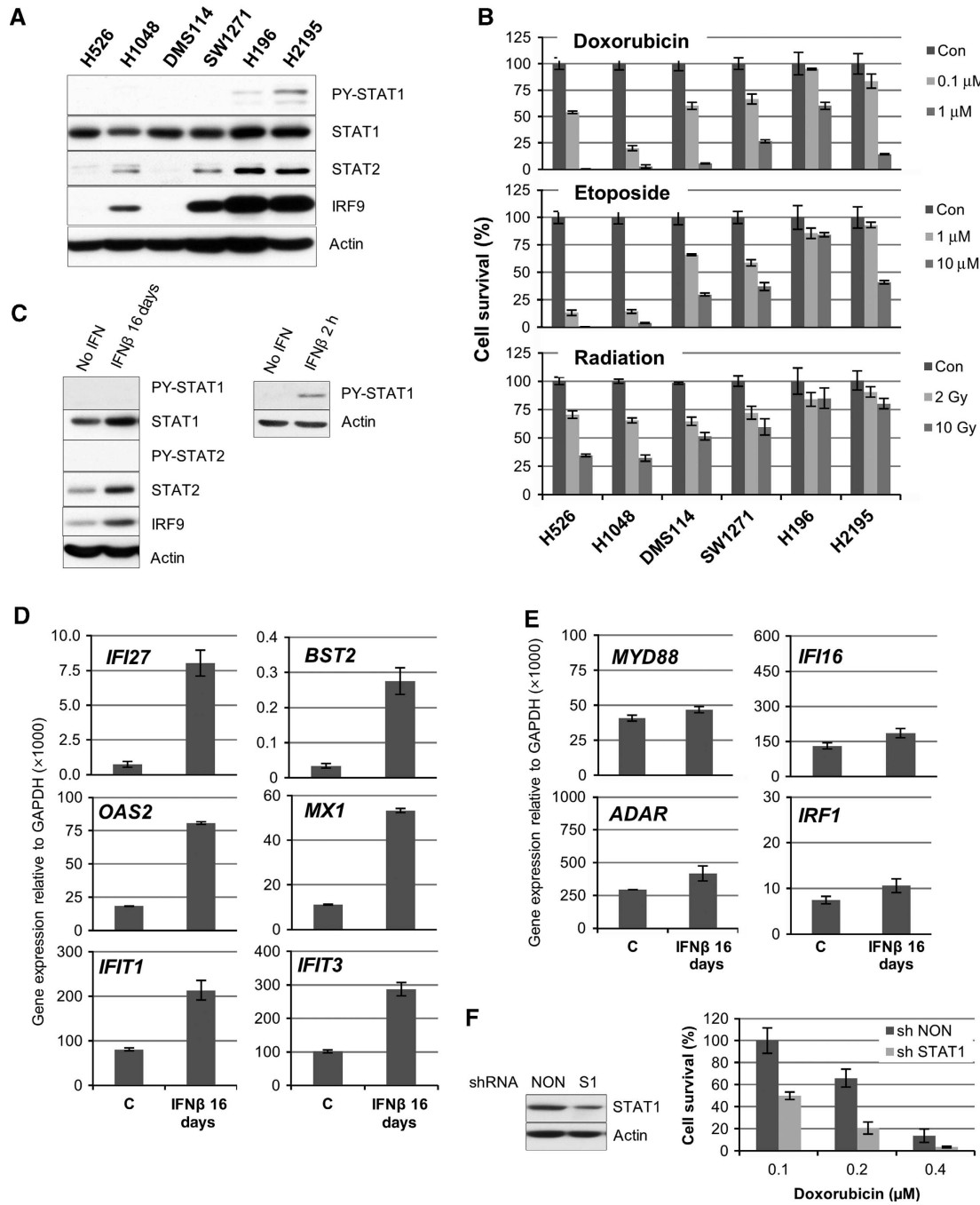

**Figure 6** U-ISGF3 induced by chronic exposure to IFNβ confers resistance to DNA damage. (**A**) The expression of PY-701- or total STAT1, PY-690- or total STAT2, and total IRF9 was examined by the western method in six different small cell lung carcinoma cell lines. A signal for PY-690-STAT2 was not detected in any lane. (**B**) The six cell lines were treated with doxorubicin, etoposide, or radiation at the indicated doses. After 72 h, cell viability was assessed by using an Alamar Blue assay. Cell numbers were determined by generating standard curves with known numbers of untreated cells. The values are percentages of the number of treated cells compared to untreated cells. The data are represented as means of triplicate Alamar Blue assays ± s.d. (**C**) BJ cells were treated with 0.5 U/ml of IFNβ every other day for 16 days (left panel) or 2 h (right panel), and the levels of total and tyrosine-phosphorylated proteins (PY-701-STAT1 or PY-690-STAT2) were examined by the western method. (**D, E**) Total RNA was extracted from BJ cells treated with 0.5 IU/ml of IFNβ every other day for 16 days. The expression of U-ISGF3-induced genes (*IFI27, BST2, OAS2, MX1, IFIT1*, and *IFIT3*) and other ISGs (*MYD88, IFI16, ADAR*, and *IRF1*) was examined by real-time PCR. The levels of gene expression were calculated semi-quantitatively by using the ΔΔCt method. The data are represented as means of triplicate PCR analyses ± s.d. (**F**) BJ cells were infected with an shRNA targeting STAT1 (S1) or a non-targeted shRNA (NON) and the levels of STAT1 protein expression were examined by the western method. The cells were treated with doxorubicin and their viability was assessed after 72 h by using an Alamar Blue assay. Cell numbers were determined by generating standard curves with known numbers of untreated cells. The values are percentages of the numbers of treated cells compared to untreated cells. The data are represented as means of triplicate of Alamar Blue assays ± s.d. Source data for this figure is available on the online supplementary information page.

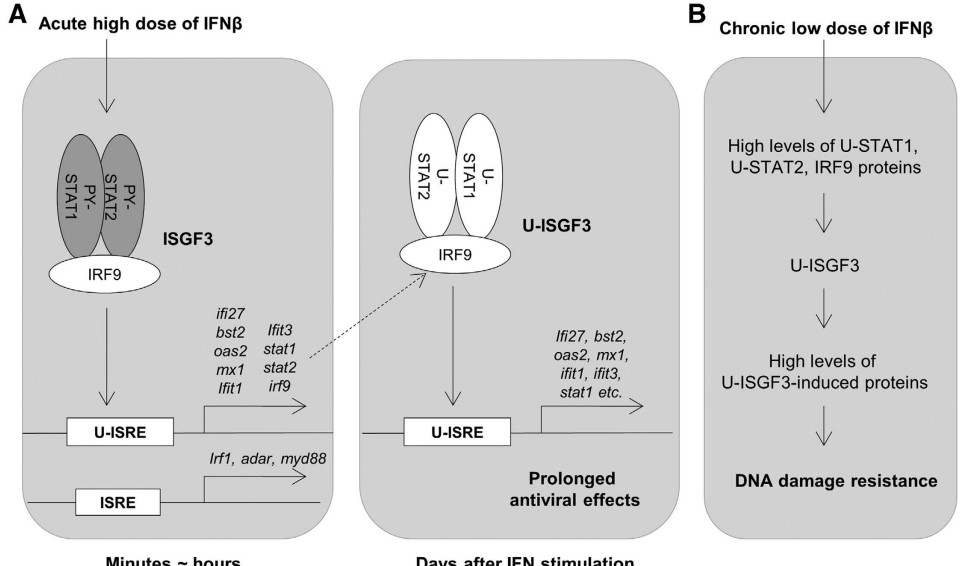

**Figure 7** Biological significance of U-ISGF3-induced gene expression. (**A**) U-ISGF3 prolongs anti-viral effects. IFNβ induces the expression of a large number of ISGs (>100 genes) within minutes to hours through the action of ISGF3 (PY-701-STAT1, PY-690-STAT2, and IRF9), which binds to standard ISREs in ISG promoters. ISGs induced by an initial treatment with IFNβ include *STAT1, STAT2,* and *IRF9,* and the encoded proteins accumulate in their unphosphorylated forms for days after IFN stimulation. The accumulated STAT1, STAT2, and IRF9 proteins form U-ISGF3, which selectively binds to the distinct ISREs in promoters of a subset of ISGs (about 30 genes), most of which are anti-viral genes. The expression of these genes is prolonged, whereas the expression of the genes induced by ISGF3, but not by U-ISGF3, is terminated rapidly. (**B**) U-ISGF3 induces resistance to DNA damage. Chronic exposure to low doses of IFNβ increases the levels of U-ISGF3 and the U-ISGF3-induced proteins, with no increase in STAT phosphorylation. The proteins induced by U-ISGF3 increase resistance to DNA damage.

addition to the differences in ISREs could be responsible for the induction of the U-ISGF3 target genes.

The long-lasting anti-viral gene expression mediated by U-ISGF3 helps to overcome countermeasures that many viruses have evolved against IFN-dependent signalling. Viruses minimize the inhibitory effects of the IFN system in many ways, for example, by decreasing the phosphorylation of STATs or by suppressing IFN synthesis (Randall and Goodbourn, 2008). Hepatitis C virus (HCV) and Japanese encephalitis virus dephosphorylate STATs through up-regulation of phosphatases (Duong *et al*, 2004; Lin *et al*, 2006). Ebola virus, Herpes simplex virus (HSV), respiratory syncytial virus, and measles virus suppress IFN synthesis by sequestering dsRNA or inhibiting TLR or RIG-I signalling (Melroe *et al*, 2004; Schlender *et al*, 2005; Cardenas *et al*, 2006). We now appreciate that, in spite of virus-induced reduction in IFN synthesis, host cells can synthesize anti-viral proteins through U-ISGF3, since expression of the STAT1, STAT2, and IRF9 proteins is increased even by low concentrations of IFNs. Therefore, by a tyrosine phosphorylation-independent mechanism, host cells can maintain at least some anti-viral functions even after IFN synthesis subsides and phosphorylated signalling molecules are inactivated.

Perwitasari *et al* (2011) recently reported that STAT1 S708 phosphorylation by IKKε is essential for expression of the anti-viral genes *IFIT2* and *ADAR1*, but not the related genes *IFIT1, IFIT3,* and *ISG15 (G1P2)*. In our microarray analysis (Figure 5A), increased levels of Y701F-STAT1 in BJ cells upregulated the expression of the IKKε-independent genes *IFIT1, IFIT3,* and *G1P2* but did not change expression of the IKKε-dependent genes *ADAR* and *IFIT2*. U-ISGF3 also increased the expression of other IKKε-independent genes

(Tenoever *et al*, 2007), including *MX2, OASL, MDA5, IRF7,* and *STAT1,* in BJ cells, indicating that the induction of U-ISGF3 target genes is independent of the S708 phosphorylation of STAT1.

Figure 7B represents our working model, depicting that chronic exposure to a low dose of IFNβ increases the levels of U-ISGF3. Elevated IFN production is often observed in many pathological conditions, such as chronic inflammation and cancer, as well as in virus infections (de Visser *et al*, 2006). In cancers, IFN production might be increased by infiltrating immune cells or by the cancer cells themselves possibly through loss of p53 function (Supplementary Figure S5; Leonova *et al*, 2013). Our data show that there is a correlation between the levels of phosphorylated Y701-STAT1 and DNA damage resistance in SCLC cell lines (Figure 6A), suggesting that DNA damage-resistant cancer cells produce IFNs in sufficient quantity to induce higher levels of STAT1, STAT2, and IRF9 proteins but not enough to induce cytotoxic genes, compared to sensitive cancer cell lines or normal cells. Low levels of IFNs upregulate only the U-ISGF3-dependent subset of ISGs in DNA damage-resistant cancer cells, with no sustained increase in the expression of the ISGs that mediate the acute apoptotic, anti-proliferative, and inflammatory responses to IFN. Recent reports show that IFNγ links UV radiation to melanomagenesis (Zaidi *et al* 2011) and that low levels of STAT phosphorylation are sufficient to induce anti-viral effects, whereas higher levels are necessary for anti-proliferative effects (Levin *et al*, 2011).

A number of recent studies have revealed that an IFN-related DNA damage-resistance signature (IRDS) predicts resistance to chemotherapy and radiation therapy in breast cancer, glioblastoma, and many other cancers (Weichselbaum *et al*, 2008; Duarte *et al*, 2012). Activation

of the JAK2/STAT1 pathway, not necessarily through phosphorylated STAT1, is related to myeloproliferative neoplasms (Chen et al, 2010), and blockade of IFNγ reduced melanomagenesis (Zaidi et al, 2011). Many experimental data have shown that high expression of IFN-induced genes, including STAT1 itself, promotes tumour growth, metastasis, and resistance to chemotherapy and radiation (Khodarev et al, 2004, 2007, 2009, 2012; Rickardson et al, 2005; Roberts et al, 2005; Luszczek et al, 2010). These findings are somewhat unexpected, since IFNβ and STAT1 have been regarded as anti-tumour factors that inhibit proliferation and promote apoptosis, predominantly through the transcriptional modulation of key proteins such as IRF1, FAS, FASL, TRAIL, p21$^{waf1}$, and caspases 2, 3, and 7 (Borden et al, 2007; Kim and Lee, 2007). What explains this paradox? We have now found that the IRDS is the same as the subset of ISGs induced by U-ISGF3 and does not include many other ISGs that confer anti-proliferative or pro-apoptotic phenotypes. The functions of most of the IRDS proteins have not been fully studied in cancer, but U-ISGF3-induced mRNAs and proteins, including IFI27 (ISG12), IFITM1 (LEU13), ISG15 (G1P2) and BST2, are upregulated in various types of cancers compared to normal tissues and in metastatic or recurrent cancers compared to the original lesions (Suomela et al, 2004; Andersen and Hassel, 2006; Hatano et al, 2008; Cai et al, 2009). These observations suggest that U-ISGF3 participates in oncogenesis as well as in resistance to therapy by inducing IRDS genes. We have observed correlations between the expression levels of STAT1, STAT2, and IRF9 and cell survival in response to DNA damage. Higher levels of U-ISGF3 in SCLC cell lines correlated with increased resistance to DNA damage, and when we knocked STAT1 and IRF9 down to reduce the levels of U-ISGF3, normal BJ fibroblasts and the resistant SCLC cell line H196 became more sensitive to DNA damage induced by doxorubicin.

In summary, our results explain (1) how cells maintain the expression of a subset of anti-viral genes despite potent negative feedback mechanisms that quickly downregulate the initial response to IFNβ (Figure 7A) and (2) the mechanism through which DNA damage-resistant cancer cells constitutively upregulate the expression of only the U-ISGF3-dependent subset of ISGs, and not the full set of ISGs including anti-proliferative or pro-apoptotic genes (Figure 7B). Future work will be aimed at revealing how individual U-ISGF3-induced proteins mediate specific anti-viral activities and resistance to DNA damage. More work is also needed to provide information on the extent to which the U-ISGF3-induced anti-viral phenotype extends to a variety of different DNA and RNA viruses.

## Materials and methods

### Constructs and gene transfection
Human STAT1, STAT2, IRF9, and Y701F-STAT1 cDNAs were cloned into the lentiviral vector pLV-tetO-CMV-SV40-Puro-LoxP (details available upon request). shRNAs in the lentiviral vector pLKO against STAT1 (NM_007315.x-838s1c1, CCGGCTGGAA GATTTAC AAG ATGAACTCGA GTTCATCTTG TAAATCTTCC AGTTTTT) STAT2 (NM_005419.2-591s21c1, CCGGTGTCTT CTGCTTCCGA TAT AACTCGA GTTATATCGG AAGCAGAAGA CATTTTTG) and IRF9 (NM_006084.3-1l10s1c1, CCGGGGCCATA CTCCACAGAA TCTTACT CGA GTAAGATTCT GTGGAGTATG GCTTTTT) were obtained from Sigma-Aldrich. To produce infectious lentiviruses, each construct was transfected into 293T packaging cells using Lipofectamine Plus (Invitrogen). The supernatant media were collected 24 and 32 h

after transfection, combined, and frozen in aliquots at −70°C. The cells were infected three times for 6 h each and the infected cells were selected with 1.5 μg/ml of puromycin for >3 days. The cells were used for subsequent experiments promptly, 7 days after infection, since the levels of STAT1 and STAT2 proteins decrease gradually over time.

### Reagents
Human IFNβ was from PBL Interferon Source. Mouse monoclonal antibodies against STAT1 (BD Transduction), STAT2, IRF9 (ISGF3γ), and IRF1 (Santa Cruz Biotechnology), and rabbit polyclonal antibodies against Tyr 701-phosphorylated STAT1, Tyr 690-phosphorylated STAT2 (Cell Signaling), and STAT1 (Upstate) were used for western analyses. Rabbit polyclonal antibodies against STAT1 and STAT2 (Upstate) and IRF9 (Santa Cruz) were used for immunoprecipitations.

### Cell culture
Human normal BJ fibroblasts were grown in DMEM supplemented with 5% fetal bovine serum (FBS), 100 U/ml penicillin, and 100 μg/ml streptomycin. Human normal mammary epithelial cells hTERT-HME1 were grown in mammary epithelium growth media (MEGM) containing bovine pituitary extract, hydrocortisone, insulin, epithelial growth factor, and gentamycin/amphotericin-B (Lonza). Human AG44122 umbilical fibroblasts were grown in Eagle's MEM with Earle's salts, non-essential amino acids, and L-glutamine containing 15% FBS, 100 U/ml penicillin, and 100 μg/ml streptomycin. Human primary STAT1-null fibroblasts (Dupuis et al, 2003) were cultured in DMEM supplemented with 10% FBS, 100 U/ml penicillin, and 100 μg/ml streptomycin. Human SCLC cell lines (H526, H1048, DMS114, SW1271, H196, and H2195) were cultured in RPMI-1640 with 2 mM L-glutamine adjusted to contain 1.5 g/l sodium bicarbonate, 10 mM HEPES, 1 mM sodium pyruvate, 10% FBS, 100 U/ml penicillin, and 100 μg/ml streptomycin. Vero cells were cultured in DMEM with 10% FBS, 100 U/ml penicillin, and 100 μg/ml streptomycin.

### Real-time PCR
cDNA was synthesized from total RNA using a modified manufacturer's protocol with random hexamer and Superscript III (Invitrogen). Real-time PCR was performed with SYBR Green qPCR master mix (USB) in an iCycler iQ real-time PCR detection system (Biorad). The PCR protocol is initial activation at 95°C for 5 min, 40 cycles at 95°C for 15 s and 60°C for 1 min. Ct values were converted into relative gene expression levels compared to that of internal control gene, GAPDH, using the ΔΔCt method or the standard curve method (Livak and Schmittgen, 2001). Each PCR run also included non-template controls containing all reagents except cDNA, which generated no amplification. The specificity was confirmed by analysis of the melting curves of the PCR products. The primer sequences are as follows:
IFI27-F, GCCTCTGGCTCTGCCGTAGTT; IFI27-R, ATGGAGGAC GAGGCGATTCC; BST2-F, ACGCGTCTGCAGAGGTGGAG; BST2-R, GCAGCGGAGCTGGAGTCCT; OAS1-F, TGAGGTCCAGGCTCCACGCT; OAS1-R, GCAGGTCGGTGCACTCCTCG; OAS2-F, AGGTGGCTCCTAT GGACGGAA; OAS2-R, GGCTTCTCTTCTGATCCTGGAATTG, MX1-F, CTTTCCAGTCCAGCTCGGCA; MX1-R, AGCTGCTGGCCGTACGTCTG; IFIT1-F, TCTCAGAGGAGCCTGGCTAA; IFIT1-R, CCAGACTATCCTTG ACCTGATGA; IFIT3-F, CAGAACTGCAGGGAAACAGC; IFIT3-R, TGA ATAAGTTCCAGGTGAAATGGC; MYD88-F, GTCTGACCGCGATGTCCT GCC; MYD88-R, ACAACCACCACCATCCGGCG; ADAR-F, ACCTGAAC ACCAACCCTGTG; ADAR-R, CGACCCCCAACTTTTGCTTG; IRF1-F, CATGAGACCCTGGCTAGAGATG; IRF1-R, TCCGGAACAAACAGGCA TCC; IFI16-F, CTACTTCACCTGCACCCTCC; and IFI16-R, TGGCCACT GTTTTTCGGGTT.

### Western analyses
Cells were resuspended in lysis buffer (250 mM Tris, pH 8.0, 150 mM NaCl, 1% Triton, 0.1% SDS) containing protease inhibitors (1 mM phenylmethanesulfonylfluoride (PMSF), 100 μg/ml aprotinin, and 1 μg/ml leupeptin) and phosphatase inhibitors (10 mM sodium fluoride, 5 mM sodium pyrophosphate, and 1 mM sodium ortho vanadate). After incubation on ice for 10–20 min, cell debris was removed by centrifugation. Protein (10–30 μg) was loaded onto 8–10% SDS–PAGE gels. The separated proteins were transferred onto polyvinylidene fluoride (PVDF) membranes (Millipore), which were incubated with primary antibody for 1–2 h, followed by incuba-

tion with horseradish peroxide (HRP)-conjugated secondary antibody for 1 h at room temperature.

### Microarray analysis

Total RNA from BJ cells treated with IFNβ for 6 h or BJ cells transfected with empty vector or Y701F-STAT1 was purified with the Trizol (Invitrogen) and RNeasy Mini Kit (Qiagen), and 1 μg of this RNA was used for microarray analysis on an Illumina Sentrix Human Ref-8 Expression Bead Chip. The analysis was carried out in duplicate with each kind of RNA and the mean of the duplicates was used for further analysis. The data were normalized by the quantile method and differential expression analysis was run with references of vector-transfected cells or untreated cells. Genes were selected according to the criteria of differential *P*-values ≤0.05 compared to control (untreated cells or empty vector-transfected cells) and average signals >25 in treated cells. The genes that increased by >2-fold were scored as being induced. Microarray data have been deposited in NCBI GEO. The accession number is GSE50954.

### Cytoplasmic-nuclear fractionation and co-immunoprecipitation

Cells were resuspended in a cytoplasm extraction buffer containing 10 mM HEPES (pH 7.9), 10 mM KCl, 0.1 mM EDTA, 0.1 mM EGTA, with proteinase/phosphatase inhibitors (10 mM sodium fluoride, 5 mM sodium pyrophosphate, 1 mM sodium vanadate, 5 μg/ml leupeptin, 1 mM PMSF, 100 μg/ml aprotinin), and exposed to 0.58% NP-40 for 5 s. After centrifugation for 30 s, the pellets were suspended in a nuclear extraction buffer containing 20 mM Tris (pH 7.5), 150 mM NaCl, 1 mM EDTA, and 1% Triton X-100 with proteinase/phosphatase inhibitors. Nuclear lysates (0.5–1 mg) were incubated with 6 μg of rabbit polyclonal antibodies overnight in the presence of 150 mM NaCl. The antibody-bound proteins were precipitated with Protein A/G PLUS Agarose (Santa Cruz), washed with PBS, and eluted with Laemmli's buffer containing 0.9% β-mercaptoethanol.

### Chromatin immunoprecipitation

This assay was performed by using the EZ ChIP kit (Upstate). After cross-linking with 1% formaldehyde and quenching in the cell-culture dishes, 9 million cells were lysed in 0.3 ml of 0.2 M triethanolamine (pH 8.0) containing 1% SDS and 0.5 μg/ml dimethyl-3′, 3′-dithiobispropionimidate (DTBP, Thermo Scientific), and incubated for 30 min at room temperature for protein-protein cross-linking. After quenching the unreacted DTBP with 50 mM glycine for 10 min, Tris/SDS lysis buffer (provided by the EZ ChIP kit manufacturer) was added to bring the total volume to 0.5 ml. Chromatin was sheared at 15% power for 38 cycles of 10 s pulses, using a Misonix Sonicator 3000. The following steps were done according to the manufacturer's protocol, with rabbit non-immune IgG or rabbit polyclonal antibodies against STAT1, STAT2, and IRF9. The quantity of DNA of interest was measured by real-time PCR in immune-precipitated chromatin samples and 2% input samples. The percentages of DNA amount in immune-precipitated samples compared to 2% input were presented (% input). The sequences of primers spanning ISREs on the gene promoter are shown below.

ChIP-IFI27-F, CTTCTGGACTGCGCATGAGG; ChIP-IFI27-R, CCACC CCGACTGAAGCACTG; ChIP-OAS2-F, CGCTGCAGTGGGTGGAGAGA; ChIP-OAS2-R, GCCGGCAAGACAGTGAATGG; ChIP-MX1-F, GGGACA GGCATCAACAAAGCC; ChIP-MX1-R, GCCCTCTCTTCTTCCAGGCAAC; ChIP-MYD88-ISRE1-F, GCTGGAAGCGAAATATGCTG; ChIP-MYD88-ISRE1-R, CAAGGGAAGGTACTGTGCTGA; ChIP-MYD88-ISRE2-F, CAC ACCTGTGGGTATTTCTCG; ChIP-MYD88-ISRE2-R, TCAATTGCTGAGC ACCAGTC; ChIP-IRF1-F, AACCCAGGCTTTCAGACTCA; ChIP-IRF1-R, TCCAACCTTCTGTCCCTGAC; ChIP-ADAR-F, AGCGGAGTGGTAAGACC AGA; ChIP-ADAR-R, GCCTGAGCTGAGACTGCAA.

### Promoter analysis

Computational promoter binding site analysis was performed on putative promoter regions 2500 bp upstream and 500 bp downstream of Ensemble (v64)-annotated start sites. Gene set enrichment analysis was performed using the CLOVER algorithm (Frith *et al*, 2004) with Transfac Professional matrices. Unguided, iterative K-mean clustering across sequence populations was performed to provide assessment of binding site similarity within sites identified with Transfac. Symmetrized, position-averaged Kullback-Leibler distance was applied to statistically quantify binding site similarity (Thijs *et al*, 2002). Transcription factor binding sites were visualized using the WebLogo tool with default settings (Crooks *et al*, 2004).

### Plaque assay

Cells were infected with EMCV or VSV in plain DMEM for 1 h, and placed in complete medium. After 6–9 h, the infected cells were frozen in cell-culture media at − 70°C. The frozen virus-infected cells were frozen and thawed twice, serially diluted ($10^{-1}$–$10^{-6}$) in plain DMEM, and placed onto 70–80% confluent Vero cells. After 1 h of incubation, 1% methyl cellulose media was added on top of the virus-infected Vero cells. After 2 days, cells with plaques were fixed with 4% paraformaldehyde (EMCV) or methanol (VSV) for at least 20 min, and stained with 1% crystal violet in 10% methanol/PBS.

### Virus infection assay using flow cytometry

At various times after infection of cells with GFP-expressing viruses (VSV, PIV3, and YFV) in 24-well plates, medium was removed and the cells were detached in Accumax Cell Aggregate Dissociation Medium (eBiosciences). The cells were fixed in 1% paraformaldehyde for at least 20 min, and stored in 1X PBS containing 3% FBS. GFP was monitored by fluorescence-activated cell sorting (FACS), using an LSRII flow cytometer (BD Biosciences).

### Cell survival assay

Cell cultures were assayed for survival using Alamar Blue (Invitrogen) according to the manufacturer's instructions. Briefly, the reagent, a vital dye that indicates mitochondrial function, was added to the cultures for >4 h and the signal was read using a fluorometer (Wallac Victor2; Perkin-Elmer).

### Supplementary data

Supplementary data are available at *The EMBO Journal* Online (http://www.embojournal.org).

## Acknowledgements

EMCV, VSV-GFP, PIV3-GFP, and YFV-GFP were kindly provided by Drs G Sen, Cleveland Clinic, J Rose, Yale University, and P Collins, National Institute of Allergy and Infectious Diseases. We thank Sen lab members, especially Jaime Wetzel and Ying Zhang, for sharing their virus experimental facility. Dr J Casanova, Rockefeller University, kindly provided human primary STAT1-null fibroblasts. The Genomics Core of the Lerner Research Institute, Cleveland Clinic, performed microarray analyses. The umbilical cord fibroblast cell line, AG14412, was provided by Dr R Padgett, Cleveland Clinic. This work was funded by National Institutes of Health Grants PO1 CA062220 to GRS, RO1 AI091707 to CMR, and K01 DK095031 to JWS. Additional funding to CMR was provided by the Greenberg Medical Research Institute and the Starr Foundation.

*Author contributions*: HC and GRS designed the research; HC, EGH-B, JWS, SF, NI, and DJJ performed the research; HC, EGH-B, JWS, SF, CMR, PH, MWJ, and GRS analysed the data; HC and GRS wrote the paper.

## Conflict of interest

The authors declare that they have no conflict of interest.

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
