## [Review Process File · The EMBO Journal]

Manuscript EMBO-2012-84264

IFN β -dependent Increases in STAT1, STAT2, and IRF9 Mediate Resistance to Viruses and DNA Damage

HyeonJoo Cheon, Elise G Holvey-Bates, John W Schoggins, Samuel Forster, Paul Hertzog, Naoko Imanaka, Charles M Rice, Mark W Jackson, Damian J Junk and George R Stark

Corresponding author: George Stark, Cleveland Clinic Foundation

Review timeline:

Submission date:	18 December 2012
Editorial Decision:	12 February 2013
Revision received:	03 July 2013
Editorial Decision:	07 August 2013
Revision received:	09 August 2013
Accepted:	13 August 2013

Transaction Report:

Editor: Karin Dumstrei

1st Editorial Decision

12 February 2013

Thank you for submitting your manuscript to the EMBO Journal. I am sorry for the delay in getting back to you with a decision, but due to the holiday season it took some additional time to get the paper reviewed. I have now heard back from the three referees and their comments are provided below.

As you can see, the referees find the characterization of the U-ISGF3 complex and its role in transcriptional regulation very interesting. However, they also raise a number of important points that should be resolved in order to consider publication here. Most of the concerns center on the U-ISGF3 complex and if there is a residual P-ISGF3 around that drives the observed transcriptional effects. They also find that further support for that ISGF3 and U-ISGF3 differentiate between the proposed binding sites is needed. Should you be able address the concerns raised then we would like to consider a revised version. I should add that it is EMBO Journal policy to allow one more round of revision only and it is therefore important to resolve the raised concerns at this stage.

When preparing your letter of response to the referees' comments, please bear in mind that this will form part of the Review Process File, and will therefore be available online to the community. For more details on our Transparent Editorial Process, please visit our website: <http://www.nature.com/emboj/about/process.html>

We generally allow three months as standard revision time. As a matter of policy, competing manuscripts published during this period will not negatively impact on our assessment of the conceptual advance presented by your study. However, we request that you contact the editor as soon as possible upon publication of any related work, to discuss how to proceed. Should you foresee a problem in meeting this three-month deadline, please let us know in advance and we may

be able to grant an extension.

Thank you for the opportunity to consider your work for publication. I look forward to your revision.

REFEREE REPORTS

Referee #1

Cheon and colleagues continue on their recent findings concerning the importance of U-Stat signaling for cellular responses to type I interferons. A combination of overexpression studies, IP and ChIP lead the authors to conclude that a U-ISGF3 complex produces transcriptional effects during the late phase of an IFN response or upon overexpression. Target genes of U-ISGF3 contain variant ISGF3 binding sites. The authors further show that cells expressing U-ISGF3 show increased antiviral activity and become resistant to DNA damaging agents. Based on their studies the authors propose that U-ISGF3 signaling prolongs the antiviral branch of the cellular response to type I IFN. They further propose that low levels of IFN production and subsequent U-ISGF3 activity produces in cancer cells a state of resistance to DNA damaging treatments.

The study presents an intriguing concept. It raises the following questions:

1. If Stat1 is U-Stat target gene shouldn't cells remain antiviral forever after a single pulse of IFN?
2. Low levels of IFN production or tonic IFN receptor signaling is thought to continuously upregulate Stats for enhanced alertness of the responding cell. The best evidence for this is the rapid drop of Stat1 levels in cell deficient for components of the type I IFN signaling pathway. What is different about cancer cells that they become resistant to DNA damage, shouldn't this happen in every cell?
3. Shouldn't low chronic IFN-beta exposure lead to pulses of Jak activity, based on the half life of the negative regulators that shut off signaling?
4. Perry et al (PLoS pathogens 7, e1001297 (2011)) convincingly show that type I IFN can produce a late response that is Stat2-dependent, but Stat1 independent. Some of the genes tested by Perry et al. (e.g. Mx, Ifit3) are U-ISGF3 targets according to this manuscript. Although the data in figure 2 suggest all ISGF3 components are needed for U-Stat signaling can the authors rule out a Stat1-independent effect on late stage IFN responses?

Specific comments:

1. The study is based exclusively on fibroblasts and tumor cells. It would be relevant to show that the U-Stat phase of IFN signaling also occurs in other cell types (e.g. epithelial cells, monocytes).
2. Figure 1/text: to my knowledge the IRF1 promoter contains a GAS, not an ISRE element.
3. Can U-ISGF3 be ChIP'ed from cells not overexpressing Stats at a late stage of the IFN response?
4. Top of p. 7: this paragraph is confusing. What is the additional effect of IFN treatment not observed in cells expressing Stat1 YF? To my opinion the data look very similar to those expressing Stat1 wt.
5. Figure 4: are the sites amplified after ChIP the same as those previously reported in the literature? Their positions should be indicated.
6. Figure 5: the term 'variant ISRE' appears somewhat discrepant to the fact that algorithms searching for optimal ISREs were used to identify them. More genes of the U-ISGF3 and non U-ISGF3 target categories should be analyzed by real-time PCR.

Referee #2

Cheon et al. present an interesting study on the molecular basis underlying the prolonged expression of a subset of type I interferon (IFN)-responsive genes. They present data describing a novel transcription complex termed U-ISGF3 and link it to this phenomenon. In contrast to conventional ISGF3, which consists of Tyr-phosphorylated (aka "activated") STAT1 and STAT2 plus IRF9, U-ISGF3 contains the unphosphorylated STATs. The formation of U-ISGF3 requires its constituents to be present at high concentrations, a situation found in multiple cancers. As certain radiation-resistant cancer cells and cells over-expressing unphosphorylated STAT1 show up-regulation of the same

subset of IFN-inducible genes, the biological activity of U-ISGF3 may be linked to radiotherapy resistance. The authors further arrive at a description of the binding motifs of ISGF3 and U-ISGF3, and conclude that they are highly related but different, thus providing a molecular explanation for the different expression profiles of ISGF3- and U-ISGF3-inducible genes. The study thus provides a provocative answer to an unsolved biological problem of considerable medical impact.

For this reviewer the basic question that requires an unambiguous answer is the following: Are the transcriptional effects observed in the presence of increased concentrations of STAT1 and STAT2 caused by the formation of U-ISGF3, or are they due to residual (phosphorylated) ISGF3? The relevance of this question is underscored by the fact that auto- and paracrine low dose interferon stimulation, which can occur in essentially all cells and which results in the Tyr-phosphorylation of STAT1 and STAT2, is the natural trigger to increase the intracellular concentrations of STATs and IRF9. The use of IFN-receptor null cells, or STAT1 or STAT2 null cells, or in fact doubly null cells that can be generated using mouse genetics, reconstituted with the respective tyrosine mutant, in this reviewer's view are the most convincing ways to clarify this issue. Cheon et al. do an experiment with U-STAT1 reconstituted fibroblast cells deficient in native STAT1, but this experiment is shown in the Supplement only. I think this is important and should be brought into the main text. The expression of additional U-ISGF3- and of ISGF3-induced genes should be tested using these cells. It is somewhat puzzling that rather than expanding upon this approach, Cheon et al. instead prefer to use cells that contain endogenous wild type STAT1 and STAT2.

In the Supplementary Figure 1 Cheon et al. also examine serine 708 phosphorylation by immunoblotting. STAT1 serine 708 phosphorylation and the antibody provided to Cheon et al. for this purpose remain poorly characterized. This aspect therefore is of little relevance for the present study in this reviewer's view. Nonetheless, Cheon et al. make the effort and investigate this point, and state that their results "confirm" earlier data, namely that serine phosphorylation of STAT1 occurs only at very late time points (SFig. 1B). The reported (weak) antibody reactivity coincides with high STAT1 amounts, which happen to occur at the late time points. Without normalization for STAT1 amounts it thus cannot be inferred that serine phosphorylation occurs at later time points only. As the previous work by others has not given this point much regard either, the timing of serine phosphorylation remains presently unknown. I do not request additional work in this direction by Cheon et al, but caution against unwarranted "confirmation" claims.

In Figures 4 and 5 the authors analyze the DNA binding of ISGF3 and U-ISGF3. In Figure 4B they show recruitment of STAT1, STAT2 and IRF9 to the promoters of U-ISGF3 target genes. The comparison with target genes of ISGF3, where U-ISGF3 should be absent (e.g. IRF1, ADAR, or MyD88), is missing. This should be included.

Figure 5 gives the binding site signatures for ISGF3 and U-ISGF3 derived from promoter analyses of genes induced by IFN β only (those are considered ISGF3 targets, n=48), or by both IFN β and STAT1 over-expression (those are considered U-ISGF3 targets, n=29). Cheon et al. assume that genes which show prolonged induction by IFN β and which are also induced by over-expressed U-STAT1 are regulated by U-ISGF3. This conviction is largely shared by this reviewer, although I think the evidence could be strengthened by the experiments described. Cheon et al then assume that U-ISGF3 binds to sites that resemble the binding sites of ISGF3. They therefore search for ISGF3-like sites and somewhat unsurprisingly find them. The rationale for this latter assumption is not stated explicitly, and I would question that this assumption is justified, as U-ISGF3 may well bind to "ISGF3-like" sites, but this does not mean these sites are physiologically relevant, or that there are not additional, possibly more important sites it can bind to. I would encourage the authors to address this point and to consider the small sample sizes. Specifically, several points are currently unclear in this regard. Firstly, 150 genes were identified that are up-regulated by IFN β . Is this the total number of up-regulated genes? If not, which selection criteria were applied? Only 29 of these 150 genes are also up-regulated in untreated cells that over-express U-STAT1. Is this the total number of up-regulated genes in those cells? In a previous experiment by Cheon et al. (2009) using the same cells, more than 100 genes were up-regulated by U-STAT1. Are there thus additional genes up-regulated by U-STAT1, which have not been included in the binding site analyses? If yes, the rationale for their exclusion should be explained, as it appears that genes that are up-regulated solely by U-STAT1 (rather than by IFN too) would be relevant for determining the binding site of U-ISGF3. It is also not explained at present why the binding motif determination for ISGF3 includes only 48 genes, rather than all 121 potential candidates (150 IFN-induced genes minus the 29 U-

ISGF3 target genes). Overall, the evidence for divergent ISGF3 and U-ISGF3 binding sites appears weak. Experimental evidence that ISGF3 and U-ISGF3 indeed differentiate between their proposed binding sites, for example EMSAs or reporter gene assays, is not included.

The paragraph starting with Khodarev et al. on page 10 is very difficult to understand and needs to be rewritten (where is the "list of DNA damage resistance genes"? how many exactly are "most" of the genes induced by U-ISGF3? what are the "15 such genes" marked with crosses shown in Figure 5a? By the way, Figure 5a has only 14 genes marked with crosses).

Figure 6 would be improved if the number of control genes tested (not induced by U-ISGF3) was larger than 2 (panel E).

Finally, in the Discussion it is stated that Figure 7 "shows" that chronic exposure to low dose IFN increases U-ISGF3. This should be worded differently as Figure 7 is a model depicting the authors' hypotheses. Also in the discussion the authors state that potent negative feedback mechanisms are at work to down-regulate ISGF3. However, whether this is indeed the case at low IFN doses and under the experimental conditions used here, must be experimentally demonstrated, but to my knowledge that has not been done to date and can be left to future work.

Referee #3

In this study, the authors suggest that some genes induced by the ISGF3 transcription complex, normally induced by IFN following tyrosine phosphorylation of STAT1 and STAT2, can also be induced by high concentrations of ISGF3 proteins in the absence of IFN and of tyrosine phosphorylation. They go on to suggest that this gene induction is important for creating a sustained antiviral response and for rendering cancer cells resistant to DNA damage. Mechanistically, they suggest that the non-phosphorylated proteins interact to form an ISGF3-like complex, and that this complex binds a distinct promoter element rather than the canonical ISGF3 target site.

These data extend previous studies by this lab and others on the nature of non-canonical STAT complexes. What seems to be missing from the current study is definitive proof that this U-ISGF3 complex is actually devoid of tyrosine phosphorylation and independent of previously described SH2 domain-dependent interactions. Two lines of evidence are presented in support of U-ISGF3. First, the inability to detect phosphorylated STAT proteins using antibody reagents, which is subject to the caveat that biologically relevant levels of phosphorylated proteins are below the limit of detection by the assay. The second line of evidence is that expression of mutant STAT1 lacking a phosphotyrosine site, along with STAT2 and IRF9, leads to increases in gene expression. However, all the over-expression studies were performed in the context of wild type endogenous STAT proteins, which obscures the interpretation. A more definitive experiment would involve sustained expression of tyrosine-mutant STAT1 and STAT2 in a cell lacking endogenous STAT proteins.

To demonstrate U-ISGF3, co-ip experiments were performed, showing interaction of STAT1, STAT2, and IRF9. It would be helpful to compare the efficiency of interaction with that observed after IFN treatment. Similarly ChIP experiments were used to measure U-ISGF3 at promoters, by no comparison was made to the efficiency of ISGF3 recruitment in response to IFN and no comparison was made between ISGF3- and U-ISGF3-responsive promoters. It would also be of interest to use mutant proteins to determine what mediates U-ISGF3 formation; e.g., is the phosphotyrosine-binding region of the SH2 domain involved?

H196 and H2195 cells were shown to be highly resistant to DNA damage and to have high levels of ISGF3 proteins. However, these cells also showed PY-STAT1, raising a question of whether ISGF3 or U-ISGF3 mediates resistance.

The only gene evaluated in the study that shows transient rather than prolonged expression in response to IFN, and therefore not a target of U-ISGF3, was IRF1. However, this gene has been described as a target for STAT1 homodimers, not ISGF3.

Response to Referee #1's questions and comments

1. If Stat1 is a U-Stat target gene shouldn't cells remain antiviral forever after a single pulse of IFN?

In our data shown in Figure S1A, we observed that the increased protein expression of STAT1, STAT2, and IRF9 is sustained up to 12 days, the longest time we could culture the cells without changing media, after a single treatment of IFN β . Accordingly, the expression of U-ISGF3-induced antiviral genes (*IFI27*, *Mx1*, and *OAS2*) also remains high for 12 days, while that of non-U-ISGF3-induced ISGs (*MyD88*, *IRF1*, and *IFI16*) returns to the basal level after 3 days or earlier (Figure S1 B and C). However, we think that the antiviral effects induced by a single pulse of IFN may not remain "forever". U-ISGF3-mediated protein expression of STAT1, STAT2, and IRF9 is not as strong as P-ISGF3-mediated expression, so the expression of ISGF3 components will gradually decrease and return to the basal level eventually. We added new data in Figure S1 and described it in the result section (p5).

2. Low levels of IFN production or tonic IFN receptor signaling is thought to continuously upregulate Stats for enhanced alertness of the responding cell. The best evidence for this is the rapid drop of Stat1 levels in cell deficient for components of the type I IFN signaling pathway. What is different about cancer cells that they become resistant to DNA damage, shouldn't this happen in every cell?

As described by Gough et al (Immunity, 2012, 36:166), tonic IFN receptor signaling by constitutive low levels of type I IFN plays a role to maintain a homeostatic balance under normal conditions. The elevated IFN production is often observed in many pathological conditions, such as chronic inflammation and cancer as well as virus infection (de Visser et al, 2006, Nature Review Cancer 6: 24). The IFN production might be increased by infiltrating immune cells around cancer cells or by cancer cells themselves possibly through loss of p53 function (Figure S5, Leonova et al, 2013, PNAS, 110: E89). Our data show that there is a correlation between the levels of phospho Y701-STAT1 and DNA damage resistance in small cell lung carcinoma cell lines (Figure 6A), suggesting DNA damage resistant cancer cells may produce more IFNs, which is enough to increase higher levels of STAT1, STAT2, and IRF9 proteins but not enough to induce cytotoxic proteins, compared to sensitive cancer cell lines or normal cells. Currently, we are investigating the detailed mechanism of how DNA damage resistant cells induce more IFN than sensitive cells. We added to the discussion section (p13).

3. Shouldn't low chronic IFN-beta exposure lead to pulses of Jak activity, based on the half life of the negative regulators that shut off signaling?

The down-regulation of Jak activity is accomplished by multiple mechanisms. SOCS1 protein binds to phosphorylated Jaks and interferon receptors, and blocks their activity. SHP-1, a tyrosine phosphatase binds to phosphorylated Jaks and STATs, and dephosphorylates those proteins (Schindler et al., 2007, JBC 282: 20059). Interferon receptors are also down-regulated by endocytosis-mediated internalization or ubiquitination after associating with ligands (Marijanovic et al, 2006, Biochem J 397: 31; Kumar KG, 2008, JBC 283: 18566). Even after negative regulators are degraded, Jaks may not be reactivated because initial signals from interferon receptors are down-regulated by internalization. In our experiments, we have not observed oscillation of STAT1 phosphorylation resulting from Jak reactivation. We did not add more data or discussion for this question since we think it is not important to the main point of this manuscript.

4. Perry et al (PLoS pathogens, 2011, 7: e1001297) convincingly show that type I IFN can produce a late response that is Stat2-dependent, but Stat1 independent. Some of the genes tested by Perry et al. (e.g. Mx, Ifit3) are U-ISGF3 targets according to this manuscript. Although the data in figure 2 suggest all ISGF3 components are needed for U-Stat signaling can the authors rule out a Stat1-independent effect on late stage IFN responses?

Our new data (Figure 1D) are critical to supporting our conclusion that the late response to IFN β is STAT1 dependent. We used STAT1-null fibroblasts reconstituted with wild-type-STAT1 in the lentiviral vector. IFN β does not increase the expression of STAT1 because the *STAT1* gene is regulated by the CMV promoter in the vector, not by the natural STAT1 promoter. At an early stage, the expression of antiviral genes is increased because the STAT1 can be phosphorylated. However, the gene expression is down-regulated at late stages even though STAT2 and IRF9 protein levels are

highly increased, showing that high levels of STAT1 protein are required to sustain high levels of antiviral genes. The authors of the paper (Perry et al PLoS pathogens, 2011, 7: e1001297) used STAT1-null mice to come to this conclusion about STAT1-independent antiviral effects at a late stage. We also observed that high levels of STAT2 and IRF9 proteins increased the expression of the same antiviral genes in STAT1-null fibroblasts, but the expression levels are much lower than that in cells expressing high STAT1 together with STAT2 and IRF9. We are investigating whether STAT2 can substitute for STAT1 in the STAT1-null background for our future publication.

5. The study is based exclusively on fibroblasts and tumor cells. It would be relevant to show that the U-Stat phase of IFN signaling also occurs in other cell types (e.g. epithelial cells, monocytes).

We performed our study using hTERT-HME1 human mammary epithelial cells (Figure 1, 2, 3, and 4) as well as fibroblasts and cancer cells. We described it in the result part (p4). We have not studied U-ISGF3-mediated signaling in monocytes, but IRF9 and unphosphorylated STATs 1 and 2 proteins are up-regulated in response to IFNs in human peripheral blood mononuclear cells and macrophages (Lethonen et al, 1997, J Immunol 159: 794), suggesting that high levels of those proteins might direct the late response to IFN in these cell types.

6. Figure 1/text: to my knowledge the IRF1 promoter contains a GAS, not an ISRE element.

IRF1 is known to be induced by IFN γ , but is also significantly up-regulated by type I IFN in many cell types we tested, including fibroblasts, epithelial cells, and many cancer cell lines. We checked the *IRF1* promoter (up to -5000 bp from the transcription starting site) using the online program TFSEARCH (<http://www.cbrc.jp/research/db/TFSEARCH.html>), and found a GAS at around -1400 bp and an ISRE at around -4000 bp. It seems to be induced by either STAT1 homo-dimer or ISGF3 depending on stimulation. Our microarray and promoter analysis revealed that 73 genes (*IRF1* is one of them) are induced by IFN β as well as IFN γ and have both STAT1 binding sites and ISRE sites in their promoters. We added this information in the result section (p9) and showed the position of the ISRE of *IRF1* in Figure S4A.

7. Can U-ISGF3 be ChIP'ed from cells not overexpressing Stats at a late stage of the IFN response?

We proved the presence of U-ISGF3 in IFN free conditions, because even a small amount of residual P-ISGF3 in IFN-treated cells might lead us to an incorrect conclusion. As an effort to respond to the reviewer's question, we ChIP'ed IRF9, the DNA binding protein of P- or U-ISGF3, in hTERT-HME1 cells treated with IFN β (3 IU/ml) for 4 or 72 hrs, and amplified the ISRE of *IFI27* gene. IRF9 did not bind to the *IFI27* promoter in untreated cells, but its binding was highly increased at 4 hrs and remained high after 72 hrs. However, PY-STAT1 binding was increased at 4 hrs, but returned to the basal level (no binding) after 72 hrs. These data indirectly show that U-ISGF3 exists at a late stage of IFN signaling. We added this data in Figure S4C and discussed it in the discussion section (p11).

8. Top of p. 7: this paragraph is confusing. What is the additional effect of IFN treatment not observed in cells expressing Stat1YF? To my opinion the data look very similar to those expressing Stat1 wt.

We restated the paragraph to avoid confusion (p7).

9. Figure 4: are the sites amplified after ChIP the same as those previously reported in the literature? Their positions should be indicated.

We marked the positions of ISREs amplified after ChIP in Figure S4A.

10. Figure 5: the term 'variant ISRE' appears somewhat discrepant to the fact that algorithms searching for optimal ISREs were used to identify them. More genes of the U-ISGF3 and non U-ISGF3 target categories should be analyzed by real-time PCR.

We agree. We changed the terminology to "distinct ISRE", "variations in ISRE" or "ISRE with variant flanking sequences" depending on the context (p9). We analyzed the expression of more U-ISGF3 genes and non-U-ISGF3 genes in hTERT-HME1 cells and STAT1-null fibroblasts overexpressing Y701F-STAT1, STAT2, and IRF9, and added the new data in Figure 2 C-F.

Response to Referee #2's questions and comments

1. Are the transcriptional effects observed in the presence of increased concentrations of STAT1 and STAT2 caused by the formation of U-ISGF3, or are they due to residual (phosphorylated) ISGF3? ... Cheon et al. do an experiment with U-STAT1 reconstituted fibroblast cells deficient in native STAT1, ... The expression of additional U-ISGF3- and of ISGF3-induced genes should be tested using these cells (STAT1-null fibroblasts).

We agree. We added substantial data using STAT1-null fibroblasts reconstituted with Y701F-STAT1 in Figure 2E-F and Figure S2. The Y701F-STAT1 reconstituted cells did not respond to IFN β (Figure S2), and high levels of Y701F-STAT1, STAT2, and IRF9 increased the expression of U-ISGF3 target genes, but not non-U-ISGF3-induced genes.

2. In the Supplementary Figure 1 Cheon et al. also examine serine 708 phosphorylation by immunoblotting. ... As the previous work by others has not given this point much regard either, the timing of serine phosphorylation remains presently unknown. I do not request additional work in this direction by Cheon et al, but caution against unwarranted "confirmation" claims.

We totally agree. Since the data regarding S708 phosphorylation have little relevance to this subject, we removed them and added discussion in the discussion section (p12).

3. In Figures 4 and 5 the authors analyze the DNA binding of ISGF3 and U-ISGF3. In Figure 4B they show recruitment of STAT1, STAT2 and IRF9 to the promoters of U-ISGF3 target genes. The comparison with target genes of ISGF3, where U-ISGF3 should be absent (e.g. IRF1, ADAR, or MyD88), is missing. This should be included.

We agree that it is a critical piece of data for this paper. We added new data showing the binding of IRF9, the DNA binding protein of U-ISGF3, to the ISREs of MyD88, IRF1, and ADAR (Figure 4C).

4-1. ...Cheon et al then assume that U-ISGF3 binds to sites that resemble the binding sites of ISGF3. They therefore search for ISGF3-like sites and somewhat unsurprisingly find them. The rationale for this latter assumption is not stated explicitly, and I would question that this assumption is justified, as U-ISGF3 may well bind to "ISGF3-like" sites, but this does not mean these sites are physiologically relevant, or that there are not additional, possibly more important sites it can bind to. I would encourage the authors to address this point and to consider the small sample sizes.

We re-stated the assumption more clearly in the text (p9). We agree that other TF binding sites and proteins, rather than the difference in ISRE alone, may be responsible for the induction of U-ISGF3 target genes. We discussed it in the discussion section (p12). We plan to investigate whether the additional binding sites are needed for U-ISGF3 gene expression.

4-2. Firstly, 150 genes were identified that are up-regulated by IFNbeta. Is this the total number of up-regulated genes? If not, which selection criteria were applied? Only 29 of these 150 genes are also up-regulated in untreated cells that over-express U-STAT1. Is this the total number of up-regulated genes in those cells? In a previous experiment by Cheon et al. (2009) using the same cells, more than 100 genes were up-regulated by U-STAT1.

A total of 150 genes are up-regulated by IFN β treatment for 6 hrs in BJ cells. We described that the expression of more than 100 genes remains high after 48 hrs of IFN simulation in our previous paper (Cheon and Stark, 2009, PNAS). It does not mean that >100 genes are up-regulated by U-STAT1. Genes other than the 29 genes might have alternative STAT1-independent secondary mechanisms to sustain their prolonged expression. We re-stated it more clearly in the results section (p9).

4-3. Are there thus additional genes up-regulated by U-STAT1, which have not been included in the binding site analyses? If yes, the rationale for their exclusion should be explained, as it appears that genes that are up-regulated solely by U-STAT1 (rather than by IFN too) would be relevant for determining the binding site of U-ISGF3.

U-STAT1 overexpression increased the expression of a total of 42 genes, but we analyzed only 29 genes that are also increased by IFN β (6 h), because we were interested in the role of U-ISGF3 in IFN β signaling. The remaining 13 genes are not increased by IFN β at late as well as early times. We re-stated the rationale more clearly in the text (p9).

4-4. It is also not explained at present why the binding motif determination for ISGF3 includes only 48 genes, rather than all 121 potential candidates (150 IFN-induced genes minus the 29 U-ISGF3 target genes).

Morrow et al. (2011, J Immunol 186: 1685) reported that IFN γ induces the expression of antiviral genes through another ISGF3-like complex, consisting of PY-STAT1, U-STAT2, and IRF9 (ISGF3 with single phosphorylation). Among 121 IFN β -induced genes, 73 genes are also induced by IFN γ (6 hrs). We assumed that these 73 genes have unique ISREs that IFN γ -related ISGF3 as well as classical ISGF3 binds, and promoter analysis proved that their ISREs are different from those of other groups. We added this information in the results section (p9).

4-5. Overall, the evidence for divergent ISGF3 and U-ISGF3 binding sites appears weak. Experimental evidence that ISGF3 and U-ISGF3 indeed differentiate between their proposed binding sites, for example EMSAs or reporter gene assays, is not included.

We added new ChIP assay data to show the differential binding of U-ISGF3 to ISREs of P- or U-ISGF3 genes (Figure 4C). While ChIP assay shows the results influenced by all molecules in cells, EMSA or reporter assay elucidates the interaction between specific cis- and trans-acting elements. In case other factors are involved in their binding, EMSA and reporter assay are not appropriate methods. We discussed the possibility that other factors in addition to the ISRE sequences might be involved in the differential binding of various ISGF3 complexes (p11).

5. The paragraph starting with Khodarev et al. on page 10 is very difficult to understand and needs to be rewritten (where is the "list of DNA damage resistance genes"? how many exactly are "most" of the genes induced by U-ISGF3? what are the "15 such genes" marked with crosses shown in Figure 5a? By the way, Figure 5a has only 14 genes marked with crosses).

We rewrote this section to clarify confusing points (p10).

6. Figure 6 would be improved if the number of control genes tested (not induced by U-ISGF3) was larger than 2 (panel E).

We analyzed the expression of 2 more non-U-ISGF3 genes (*IFI16* and *IRF1*) in BJ cells treated with IFN β for 16 days and added the data in Figure 6E.

7. Finally, in the Discussion it is stated that Figure 7 "shows" that chronic exposure to low dose IFN increases U-ISGF3. This should be worded differently as Figure 7 is a model depicting the authors' hypotheses. Also in the discussion the authors state that potent negative feedback mechanisms are at work to down-regulate ISGF3. However, whether this is indeed the case at low IFN doses and under the experimental conditions used here, must be experimentally demonstrated, but to my knowledge that has not been done to date and can be left to future work.

We made the correction (p11). The detailed molecular mechanism of how chronic exposure to IFN increases only U-ISGF3 will be studied in the future. Thank you for the suggestion.

Response to Referee #3's comments

1. ... all the over-expression studies were performed in the context of wild type endogenous STAT proteins, which obscures the interpretation. A more definitive experiment would involve sustained expression of tyrosine-mutant STAT1 and STAT2 in a cell lacking endogenous STAT proteins.

We added more data obtained using STAT1-null fibroblasts reconstituted with Y701F-STAT1 in Figure 2E-F and Figure S2. Y701F-STAT1 reconstituted cells did not respond to IFN β (Figure S2), and high levels of Y701F-STAT1 together with STAT2 and IRF9 increased the expression of U-ISGF3 target genes, but not P-ISGF3-induced genes.

2-1. To demonstrate U-ISGF3, co-ip experiments were performed, showing interaction of STAT1, STAT2, and IRF9. It would be helpful to compare the efficiency of interaction with that observed after IFN treatment. Similarly ChIP experiments were used to measure U-ISGF3 at promoters, but no comparison was made to the efficiency of ISGF3 recruitment in response to IFN and no comparison was made between ISGF3- and U-ISGF3-responsive promoters.

We used the overexpression system to prove the existence of U-ISGF3 without influence of P-STATs. IFN β treatment increased the binding of IRF9, the DNA binding component of ISGF3, to the *IFI27* ISRE by about 40 fold compared to untreated cells) after 4 hrs. The interaction was not

decreased until 72 hrs. The IRF9 binding was 4 fold higher in cells overexpressing STAT1/STAT2/IRF9 proteins compared to empty vector-transfected control cells. We added new data in Figure S4C, but, we think that this kind of comparison is not useful, because the IFN-treated system will show data resulting from a mixture of P-ISGF3 and U-ISGF3. The levels of ISGF3 proteins are also different between the overexpression system and the IFN-treated system, and U-ISGF3 binding efficiency to the ISRE is influenced by the ISGF3 protein levels (Figure S4A). The binding of U-ISGF3 on ISGF3-responsive promoters (*MyD88*, *IRF1*, and *ADAR1*) was examined and the new data are added in Figure 4C.

2-2. It would also be of interest to use mutant proteins to determine what mediates U-ISGF3 formation; e.g., is the phosphor tyrosine-binding region of the SH2 domain involved?

Since the SH2 domain is important to stabilize P-ISGF3, it is a valuable experiment to investigate whether the SH2 domain is also involved in U-ISGF3 formation without phosphorylation. We appreciate the valuable suggestion, but the main topic of this paper is to prove the existence of U-ISGF3 and its biological functions. We plan to study more detail about U-ISGF3 and its binding sites in the future.

3. H196 and H2195 cells were shown to be highly resistant to DNA damage and to have high levels of ISGF3 proteins. However, these cells also showed PY-STAT1, raising a question of whether ISGF3 or U-ISGF3 mediates resistance.

We hypothesize that IFNs are the factors to increase U-ISGF3 in DNA damage-resistant cancer cells. It is hard to distinguish the effects of U-ISGF3 from those of ISGF3 in the presence of IFN, and we think that ISGF3 and U-ISGF3 cooperate to induce DNA damage resistance as long as the ISGF3 signal is not strong enough to induce cytotoxic genes, because the DNA damage resistance genes are induced by ISGF3 as well as U-ISGF3, and ISGF3 increases U-ISGF3 levels, too. We added it in the discussion section (p13).

4. The only gene evaluated in the study that shows transient rather than prolonged expression in response to IFN, and therefore not a target of U-ISGF3, was IRF1. However, this gene has been described as a target for STAT1 homodimers, not ISGF3.

IRF1 is known to be induced by IFN γ , but it is also induced by type I IFN in many cell types we tested, including fibroblasts, epithelial cells, and many cancer cells. We checked the *IRF1* promoter (up to -5000 bp from transcription starting site) using the online program TFSERCH (<http://www.cbrc.jp/research/db/TFSEARCH.html>), and found a GAS at around -1400 bp and an ISRE at around -4000 bp (Figure S4A). It seems to be induced by either STAT1 homo-dimer or ISGF3 depending on stimulation. Our microarray and promoter analysis revealed that 73 genes (*IRF1* is one of them) are induced by both IFN β and IFN γ and have both STAT1 binding sites and ISRE sites in the promoter. We added this information in the result section (p9) and showed the position of the ISRE of *IRF1* in Figure S4A.

2nd Editorial Decision

07 August 2013

Thank you for submitting your revised manuscript to The EMBO Journal. Your study has now been re-reviewed by referees #1 and 2. As you can see below, both referees appreciate the introduced changes and support publication here. Referee #1 has one remaining suggestion regarding supplemental figure 4C. The suggested experiment should be straightforward to do. Let me know if this is not the case.

Once we get the revised version back we will proceed with its acceptance for publication here.

REFEREE REPORTS

Referee #1

The authors' response to my comments is adequate with one exception. The experiment shown in figure S4C is not an optimal approach to address the question whether U-ISGF3 can be found at

selected promoters at the late stage of the IFN response. The figure shows that the ChIP with the Ig control varies, but the Stat1pY stays constant. To my opinion a better experiment would demonstrate that a Stat1 ChIP can be re-ChIPed with a Stat1pY antibody in the early phase, but not in the late phase.

Referee #2

The authors have satisfactorily addressed the points raised previously.

2nd Revision - authors' response

09 August 2013

Point by point response to Referee's comments

“IFN β -dependent Increases in STAT1, STAT2, and IRF9 Mediate Resistance to Viruses and DNA Damage” by Cheon et al.

Thanks to the referees for reviewing our revised manuscript, and for the positive responses. We would like to discuss Referee #1's new comment, since there seems to be a misunderstanding of the data in Figure S4C.

The referee said that the ChIP values for the IgG controls vary, but the PY-STAT1 values stay constant. However, this is not what is shown in the figure.

1. The ChIP assay needs an IgG control to measure the background non-specific signal. In every experiment, we always see background signals that differ from sample to sample (see other graphs of our ChIP data), because of subtle differences in the amounts of lysate, agarose beads, etc.
2. The PY-STAT1 ChIP signal is NOT different from the IgG background signal in the IFN-untreated sample (C) and the 72h-treated sample, showing that there is no specific PY-STAT1 binding at these time points. In the 4h-treated sample, the PY-STAT1 ChIP signal is significantly higher than the IgG background signal, showing that PY-STAT1 binds to the promoter at this time point.

The referee suggested that we should re-ChIP a Stat1 ChIP with a PY-Stat1 antibody. However, we don't understand what new information we would get through this experiment since we already know that PY-STAT1 is not present on the promoter at 72 h from the experiment shown and, furthermore, there is little if any PY-STAT1 present in the cell at this time.

We ask that *EMBO Journal* accepts the revision of the paper as is, without requiring us to perform the additional experiment requested by Reviewer #1.